# On the Optimal Weighted $\ell_2$ Regularization in Overparameterized Linear Regression

**Denny Wu**[*]
University of Toronto and Vector Institute
dennywu@cs.toronto.edu

**Ji Xu**[*]
Columbia University
jixu@cs.columbia.edu

## Abstract

We consider the linear model $\boldsymbol{y} = \boldsymbol{X}\boldsymbol{\beta}_\star + \boldsymbol{\epsilon}$ with $\boldsymbol{X} \in \mathbb{R}^{n \times p}$ in the overparameterized regime $p > n$. We estimate $\boldsymbol{\beta}_\star$ via generalized (weighted) ridge regression: $\hat{\boldsymbol{\beta}}_\lambda = (\boldsymbol{X}^\top \boldsymbol{X} + \lambda \boldsymbol{\Sigma}_w)^\dagger \boldsymbol{X}^\top \boldsymbol{y}$, where $\boldsymbol{\Sigma}_w$ is the weighting matrix. Under a random design setting with general data covariance $\boldsymbol{\Sigma}_x$ and general prior on the true coefficients $\mathbb{E}\boldsymbol{\beta}_\star \boldsymbol{\beta}_\star^\top = \boldsymbol{\Sigma}_\beta$, we provide an exact characterization of the prediction risk $\mathbb{E}(y - \boldsymbol{x}^\top \hat{\boldsymbol{\beta}}_\lambda)^2$ in the proportional asymptotic limit $p/n \to \gamma \in (1, \infty)$. Our general setup leads to a number of interesting findings. We outline precise conditions that decide the sign of the optimal choice $\lambda_{\text{opt}}$ of the ridge parameter $\lambda$, based on the *alignment* between $\boldsymbol{\Sigma}_x$ and $\boldsymbol{\Sigma}_\beta$; this rigorously justifies the surprising empirical observation that $\lambda_{\text{opt}}$ can be *negative* in the overparameterized regime. We also discuss the risk monotonicity of optimally tuned ridge regression, and confirm the double descent phenomenon for principal component regression (PCR) under anisotropic $\boldsymbol{X}$ and $\boldsymbol{\beta}_\star$. Finally, we determine the optimal $\boldsymbol{\Sigma}_w$ for both the ridgeless ($\lambda \to 0$) and optimally regularized ($\lambda = \lambda_{\text{opt}}$) case, and demonstrate the advantage of the weighted objective over standard ridge regression and PCR.

## 1 Introduction

In this work we consider learning the target signal $\boldsymbol{\beta}_\star$ in the following linear regression model:

$$y_i = \boldsymbol{x}_i^\top \boldsymbol{\beta}_\star + \epsilon_i, \quad i = 1, 2, \ldots, n$$

where each feature vector $\boldsymbol{x}_i \in \mathbb{R}^p$ and noise $\epsilon_i \in \mathbb{R}$ are drawn i.i.d. from the two independent random variables $\tilde{x}$ and $\tilde{\epsilon}$ satisfying $\mathbb{E}\tilde{\epsilon} = 0$, $\mathbb{E}\tilde{\epsilon}^2 = \tilde{\sigma}^2$, $\tilde{\boldsymbol{x}} = \boldsymbol{\Sigma}_x^{1/2} \boldsymbol{z}/\sqrt{n}$, and the components of $\boldsymbol{z}$ are i.i.d. random variables with zero mean, unit variance, and bounded 12th absolute central moment. To estimate $\boldsymbol{\beta}_\star$ from $(\boldsymbol{x}_i, y_i)$, we consider the following generalized ridge regression estimator:

$$\hat{\boldsymbol{\beta}}_\lambda = (\boldsymbol{X}^\top \boldsymbol{X} + \lambda \boldsymbol{\Sigma}_w)^\dagger \boldsymbol{X}^\top \boldsymbol{y}, \tag{1.1}$$

in which $\boldsymbol{X} \in \mathbb{R}^{n \times p}$ is the feature matrix, $\boldsymbol{y}$ is vector of the observations, $\boldsymbol{\Sigma}_w$ is a positive definite weighting matrix, and the symbol $\dagger$ denotes the Moore-Penrose pseudo-inverse. When $\lambda \geq 0$, $\hat{\boldsymbol{\beta}}_\lambda$ minimizes the squared loss plus a weighted $\ell_2$ regularization: $\min_{\boldsymbol{\beta}} \sum_{i=1}^n (y_i - \boldsymbol{x}_i^\top \boldsymbol{\beta})^2 + \lambda \boldsymbol{\beta}^\top \boldsymbol{\Sigma}_w \boldsymbol{\beta}$. Note that $\boldsymbol{\Sigma}_w = \boldsymbol{I}_d$ reduces the objective to standard ridge regression.

While the standard ridge regression estimator is relatively well-understood in the data-abundant regime ($n > p$), several interesting properties have been recently discovered in high dimensions, especially when $p > n$. For instance, the double descent phenomenon suggests that overparameterization may not result in overfitting due to the *implicit regularization* of the least squares estimator [HMRT19, BLLT19]. This implicit regularization also relates to the surprising empirical finding that the optimal ridge parameter $\lambda$ can be negative in the overparameterized regime [KLS20].

---

[*]Equal contribution; alphabetical ordering.

Motivated by the observations above, we analyze the estimator $\hat{\boldsymbol{\beta}}_\lambda$ in the proportional limit: $p/n \to \gamma \in (1, \infty)$[2] as $n, p \to \infty$. We place a general prior on the true parameters (independent of $\tilde{\boldsymbol{x}}$ and $\tilde{\epsilon}$): $\mathbb{E}\boldsymbol{\beta}_\star\boldsymbol{\beta}_\star^\top = \boldsymbol{\Sigma}_\beta$, which covers both *random* and *deterministic* $\boldsymbol{\beta}_*$. Our goal is to study the prediction risk of $\hat{\boldsymbol{\beta}}_\lambda$: $\mathbb{E}_{\tilde{x}, \tilde{\epsilon}, \boldsymbol{\beta}_\star}(\tilde{y} - \tilde{\boldsymbol{x}}^\top\hat{\boldsymbol{\beta}}_\lambda)^2$, where $\tilde{y} = \tilde{\boldsymbol{x}}^\top\boldsymbol{\beta}_\star + \tilde{\epsilon}$[3]. Compared to previous high-dimensional analysis of ridge regression [DW18], our setup is generalized in two important aspects:

**Anisotropic $\boldsymbol{\Sigma}_x$ and $\boldsymbol{\Sigma}_\beta$.** Our analysis handles general prior $\boldsymbol{\Sigma}_\beta$ and data covariance $\boldsymbol{\Sigma}_x$, in contrast to previous works which assume either isotropic features or signal (e.g., [DW18, HMRT19]). Note that the isotropic assumption on the signal or features implies that each component is roughly of the same magnitude, which may not hold true in practice. For instance, the optimal ridge penalty is *provably* non-negative when either the signal $\boldsymbol{\Sigma}_\beta$ [DW18, Theorem 2.1] or the features $\boldsymbol{\Sigma}_x$ [HMRT19, Theorem 5] is isotropic. On the other hand, it has been empirically demonstrated that the optimal ridge for real-world data can be negative [KLS20]. While this observation cannot be captured by previous works, our less restrictive assumptions lead to a concise description of this phenomenon.

**Weighted $\ell_2$ Regularization.** We consider generalized ridge regression instead of simple isotropic shrinkage. While the generalized formulation has also been studied (e.g., [HK70, Cas80]), to the best of our knowledge, no existing work computes the exact risk in the overparameterized proportional limit and decides the corresponding optimal $\boldsymbol{\Sigma}_w$. Our setting is also inspired by recent observations in deep learning that weighted $\ell_2$ regularization often achieves better generalization compare to isotropic weight decay [ZWXG18]. Our analysis illustrates the benefit of weighted $\ell_2$ regularization.

Under the general setup (1.1), the contributions of this work can be summarized as (see Figure 1):

- **Exact Asymptotic Risk.** In Section 4 we derive the prediction risk $R(\lambda)$ of our estimator (1.1) in its bias-variance decomposition (see Figure 2). We also characterize principal component regression (PCR) and confirm the double descent phenomenon under more general setting than [XH19].

- **"Negative Ridge" Phenomenon.** In Section 5, we analyze the optimal regularization strength $\lambda_{\text{opt}}$ under different $\boldsymbol{\Sigma}_w$, and provide precise conditions under which the optimal $\lambda_{\text{opt}}$ is negative in the overparameterized regime. In brief, we show that $\lambda_{\text{opt}}$ is negative when SNR is large and the large directions of $\boldsymbol{\Sigma}_x$ and $\boldsymbol{\beta}_\star$ are *aligned* (see Figure 4), and vice versa. On the other hand, we show that the optimal ridge penalty is always non-negative in the underparameterized regime ($p < n$); this implies an implicit $\ell_2$ regularization effect of overparameterization for certain cases.

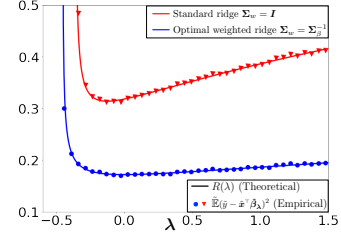

- **Optimal Weighting Matrix $\boldsymbol{\Sigma}_w$.** In Section 6, we decide the optimal $\boldsymbol{\Sigma}_w$ for both the optimally regularized ridge estimator ($\lambda = \lambda_{\text{opt}}$) and the ridgeless limit ($\lambda \to 0$). In the ridgeless limit, based on the bias-variance decomposition, we show that the optimal $\boldsymbol{\Sigma}_w$ should interpolate between $\boldsymbol{\Sigma}_x$, which minimizes the variance, and $\boldsymbol{\Sigma}_\beta^{-1}$, which minimizes the bias. Whereas for the optimally regularized case, in many settings the optimal $\boldsymbol{\Sigma}_w$ is simply $\boldsymbol{\Sigma}_\beta^{-1}$ (Figure 6) (for more general result see Theorem 10). We demonstrate the benefit of weighted regularization over standard ridge regression and PCR, and also propose a heuristic choice of $\boldsymbol{\Sigma}_w$ when information of $\boldsymbol{\beta}_\star$ is not present.

Figure 1: Illustration of the "negative ridge" phenomenon and the advantage of weighted $\ell_2$ regularization under "aligned" $\boldsymbol{\Sigma}_x$ and $\boldsymbol{\Sigma}_\beta$. We set $\gamma = 2$, $\tilde{\sigma}^2 = 0$. Red: standard ridge regression ($\boldsymbol{\Sigma}_w = \boldsymbol{I}$); note that the lowest prediction risk is achieved when $\lambda < 0$. Blue: optimally weighted ridge regression ($\boldsymbol{\Sigma}_w = \boldsymbol{\Sigma}_\beta^{-1}$), which achieves lower risk compared to the isotropic shrinkage.

**Notations:** We denote $\tilde{\mathbb{E}}$ as taking expectation over $\boldsymbol{\beta}_\star, \tilde{x}, \tilde{\epsilon}$. Let $\boldsymbol{d}_x, \boldsymbol{d}_\beta, \boldsymbol{d}_w$ be the vectors of the eigenvalues of $\boldsymbol{\Sigma}_x, \boldsymbol{\Sigma}_\beta$ and $\boldsymbol{\Sigma}_w$ respectively. We use $\mathbb{I}_S$ as the indicator function of set $S$. We write $\xi = \mathbb{E}(\tilde{\boldsymbol{x}}^\top\boldsymbol{\beta}_\star)^2/(\gamma\tilde{\sigma}^2)$ as the signal-to-noise ratio (SNR) of the problem.

## 2 Related Works

**Asymptotics of Ridge Regression.** The prediction risk of standard ridge regression ($\boldsymbol{\Sigma}_w = \boldsymbol{I}_d$) in the proportional asymptotics has been widely studied. When the data is isotropic, precise characterization can be obtained from random matrix theory [Kar13, Dic16, HMRT19], approximate message

passing algorithm [DM16], or the convex Gaussian min-max theorem[4][TAH18]. Under general data covariance, closely related to our work is [DW18], which considered a random effects model with isotropic prior on the target coefficients ($\boldsymbol{\Sigma}_\beta = \boldsymbol{I}_d$). Our risk characterization is built upon the general random matrix result of [RM11, LP11]. Similar tools have been applied in the analysis of sketching [LD19] and the connection between ridge regression and early stopping [AKT19, Lol20].

**Weighted Regularization.** The formulation (1.1) was first introduced in [HK70], and many choices of weighting matrix have been proposed [Str78, Cas80, MS05, MS18]; yet since these estimators are usually derived in the $n > p$ setup, their effectiveness in the high-dimensional and overparameterized regime is largely unknown. In semi-supervised linear regression, it is known that weighted matrix estimated from unlabeled data can improve the model performance [RC15, TCG20]. In deep learning, anisotropic Gaussian prior on the parameters enjoyed empirical success [LW17, ZTSG19]. Additionally, decoupled weight decay [LH17] and elastic weight consolidation [KPR+17] can both be interpreted as an $\ell_2$ regularization weighted by an approximate Fisher information matrix [ZWXG18], which relates to the Fisher-Rao norm [LPRS17]. Finally, beyond the $\ell_2$ penalty, weighted regularization is also effective in LASSO regression [Zou06, CWB08, BVDBS+15].

**Benefit of Overparamterization.** Our overparameterized setting is partially motivated by the double descent phenomenon [BHMM18], which can be theoretically explained in linear regression [AS17, HMRT19, BLLT19], random features regression [MM19, dRBK20, DL20], and max-margin classification [MRSY19, DKT19, HMX20], although translation to neural networks can be more nuanced [BES+20]. For least squares regression, it has been shown in special cases that overparameterization induces an implicit $\ell_2$ regularization [KLS20, DLM19], which agrees with the absence of overfitting. This observation also leads to the speculation that the optimal ridge penalty in the overparameterized regime may be negative, to partially cancel out the implicit regularization. While the possibility of negative ridge parameter has been noted in [HG83, BS99], theoretical understanding of its benefit is largely missing, expect for heuristic argument (and empirical evidence) in [KLS20]. We provide a rigorous characterization of this "negative ridge" phenomenon.

**Concurrent Works.** Independent to our work, [RMR20] computed the asymptotic prediction risk under a similar extension of the isotropic assumption on $\boldsymbol{\Sigma}_\beta$, but did not consider the sign of $\lambda_{\mathrm{opt}}$. We note that their result requires codiagonalizability of the covariances and certain functional relations between the eigenvalues, which is more restrictive than our setting. [TB20] provided a non-asymptotic analysis of ridge regression and constructed a specific spike model[5] in which negative regularization may lead to better generalization bound than interpolation ($\lambda = 0$). In a companion work [ABG+20], we connect properties of the ridgeless limit of the generalized ridge regression estimator to the implicit bias of preconditioned gradient descent (e.g., natural gradient descent), which allows us to decide the optimal preconditioner (for generalization) in the interpolation setting.

## 3  Setup and Assumptions

In addition to the prediction risk of the weighted ridge estimator $\hat{\boldsymbol{\beta}}_\lambda = (\boldsymbol{X}^\top \boldsymbol{X} + \lambda \boldsymbol{\Sigma}_w)^\dagger \boldsymbol{X}^\top \boldsymbol{y}$, the setup of which we outlined in Section 1, we also analyze the principal component regression (PCR) estimator: for $\theta \in [0, 1]$, the PCR estimator is given as $\hat{\boldsymbol{\beta}}_\theta = (\boldsymbol{X}_\theta^\top \boldsymbol{X}_\theta)^\dagger \boldsymbol{X}_\theta^\top \boldsymbol{y}$, where $\boldsymbol{X}_\theta = \boldsymbol{X} \boldsymbol{U}_\theta$ and the columns of $\boldsymbol{U}_\theta \in \mathbb{R}^{p \times \theta p}$ are the leading $\theta p$ eigenvectors of $\boldsymbol{\Sigma}_x$.

Under the setup on $(\tilde{\boldsymbol{x}}, \boldsymbol{\beta}_\star, \tilde{\epsilon})$ described in Section 1, the prediction risk of (1.1) can be simplified as

$$\tilde{\mathbb{E}}\left(\tilde{y} - \tilde{x}^\top \hat{\boldsymbol{\beta}}_\lambda\right)^2 = \underbrace{\tilde{\sigma}^2 \left(1 + \frac{1}{n}\operatorname{tr}\left(\boldsymbol{\Sigma}_{x/w}\left(\boldsymbol{X}_{/w}^\top \boldsymbol{X}_{/w} + \lambda \boldsymbol{I}\right)^{-1} - \lambda \boldsymbol{\Sigma}_{x/w}\left(\boldsymbol{X}_{/w}^\top \boldsymbol{X}_{/w} + \lambda \boldsymbol{I}\right)^{-2}\right)\right)}_{\text{Part 1, Variance}}$$

$$+ \underbrace{\frac{\lambda^2}{n}\operatorname{tr}\left(\boldsymbol{\Sigma}_{x/w}\left(\boldsymbol{X}_{/w}^\top \boldsymbol{X}_{/w} + \lambda \boldsymbol{I}\right)^{-1}\boldsymbol{\Sigma}_{w\beta}\left(\boldsymbol{X}_{/w}^\top \boldsymbol{X}_{/w} + \lambda \boldsymbol{I}\right)^{-1}\right)}_{\text{Part 2, Bias}}, \qquad (3.1)$$

where $\boldsymbol{X}_{/w} = \boldsymbol{X}\boldsymbol{\Sigma}_w^{-1/2}, \boldsymbol{\Sigma}_{x/w} = \boldsymbol{\Sigma}_w^{-1/2}\boldsymbol{\Sigma}_x \boldsymbol{\Sigma}_w^{-1/2}, \boldsymbol{\Sigma}_{w\beta} = \boldsymbol{\Sigma}_w^{1/2}\boldsymbol{\Sigma}_\beta \boldsymbol{\Sigma}_w^{1/2}$. Note that the variance term does not depend on the true signal, and the bias is independent of the noise level. Let $\boldsymbol{d}_{x/w}$ be

the eigenvalues of $\boldsymbol{\Sigma}_{x/w}$ and $\boldsymbol{\Sigma}_{x/w} = \boldsymbol{U}_{x/w}\boldsymbol{D}_{x/w}\boldsymbol{U}_{x/w}^\top$ be the eigendecomposition of $\boldsymbol{\Sigma}_{x/w}$, where $\boldsymbol{U}_{x/w}$ is the eigenvector matrix and $\boldsymbol{D}_{x/w} = \operatorname{diag}(\boldsymbol{d}_{x/w})$. Let $\boldsymbol{d}_{w\beta} \triangleq \operatorname{diag}\left(\boldsymbol{U}_{x/w}^\top \boldsymbol{\Sigma}_{w\beta}\boldsymbol{U}_{x/w}\right)$. When $\boldsymbol{\Sigma}_w = \boldsymbol{I}$, $\boldsymbol{d}_{w\beta}$ characterizes the strength of the signal $\boldsymbol{\beta}_\star$ along the directions of the eigenvectors of feature covariance $\boldsymbol{\Sigma}_x$. To simplify the RHS of (3.1), we make the following assumption:

**Assumption 1.** *Let $d_{x/w,i}$ and $d_{w\beta,i}$ be the $i$th entry of $\boldsymbol{d}_{x/w}$ and $\boldsymbol{d}_{w\beta}$, respectively. The empirical distribution of $(d_{x/w,i}, d_{w\beta,i})$ jointly converges to non-negative random variables $(h, g)$. Further, $\min_i d_{x/w,i} \geq c_l$, $\max_i(d_{x/w,i}, d_{w\beta,i}) \leq c_u$ and $\|\boldsymbol{\Sigma}_{w\beta}\| \leq c_u$ for some $c_l, c_u > 0$ independent of $p$.*

One can check that $\boldsymbol{\Sigma}_x$ and $\boldsymbol{\Sigma}_\beta$ studied in [DW18, HMRT19, XH19] (with $\boldsymbol{\Sigma}_w = \boldsymbol{I}$) are special cases of Assumption 1 with either $h$ or $g$ being a point mass. Our Assumption 1 thus covers much more general settings of $\boldsymbol{\Sigma}_x$ and $\boldsymbol{\Sigma}_\beta$, which allows us to precisely analyze the *negative ridge* phenomenon.

## 4   Risk Characterization

With the aforementioned assumptions, we now present our characterization of the prediction risk.

**Theorem 1.** *Under Assumption 1, the asymptotic prediction risk is given as*

$$\tilde{\mathbb{E}}\left(\tilde{y} - \tilde{x}^\top \hat{\boldsymbol{\beta}}_\lambda\right)^2 \xrightarrow{p} \frac{m'(-\lambda)}{m^2(-\lambda)} \cdot \left(\gamma\mathbb{E}\frac{gh}{(h\cdot m(-\lambda)+1)^2} + \tilde{\sigma}^2\right) := R(\lambda), \ \forall \lambda > -c_0 \ \ (4.1)$$

*where $c_0 = (\sqrt{\gamma} - 1)^2 c_l$, and $m(z)$ is the Stieltjes transform of the limiting distribution of the eigenvalues of $\boldsymbol{X}_{/w}\boldsymbol{X}_{/w}^\top$. Additionally, $m(-\lambda), m'(-\lambda) > 0$ satisfy the self-consistent equations:*

$$\lambda = \frac{1}{m(-\lambda)} - \gamma\mathbb{E}\frac{h}{1 + h\cdot m(-\lambda)} \quad (4.2)$$

$$1 = \left(\frac{1}{m^2(-\lambda)} - \gamma\mathbb{E}\frac{h^2}{(h\cdot m(-\lambda)+1)^2}\right)m'(-\lambda). \quad (4.3)$$

Note that the condition $\lambda > -c_0$ ensures both $m(-\lambda)$ and $m'(-\lambda)$ exist and are positive. Furthermore, it can be shown from prior works [DW18, XH19] that the variance term (part 1) in (3.1), converges to $\tilde{\sigma}^2 \frac{m'(-\lambda)}{m^2(-\lambda)}$. Our main contribution is to characterize the bias term, Part 2, under significantly less restrictive assumption on $(\boldsymbol{\Sigma}_x, \boldsymbol{\Sigma}_\beta, \boldsymbol{\Sigma}_w)$. In particular, we show that

$$\text{Part 2} \xrightarrow{p} \frac{m'(-\lambda)}{m^2(-\lambda)} \cdot \gamma\mathbb{E}\frac{gh}{(h\cdot m(-\lambda)+1)^2}, \quad \forall \lambda > -c_0.$$

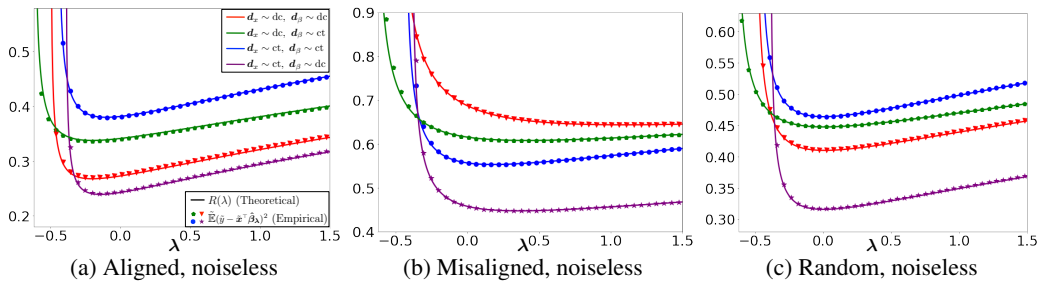

(a) Aligned, noiseless    (b) Misaligned, noiseless    (c) Random, noiseless

Figure 2: Finite sample prediction risk $\tilde{\mathbb{E}}(\tilde{y} - \tilde{\boldsymbol{x}}^\top\boldsymbol{\beta}_\star)^2$ (experiment) and the asymptotic risk $R(\lambda)$ (theory) against $\lambda$ for standard ridge regression ($\boldsymbol{\Sigma}_w = \boldsymbol{I}_d$). We set $\gamma = 2$ and $(n, p) = (300, 600)$. 'dc' and 'ct' stand for for discrete and continuous distribution, respectively. We write 'aligned' if $\boldsymbol{d}_x$ and $\boldsymbol{d}_\beta$ have the same order, 'misaligned' for the reverse, and 'random' for random order. Colors indicate different combinations of $\boldsymbol{d}_x$ and $\boldsymbol{d}_\beta$. Note that our derived risk $R(\lambda)$ matches the experimental values, and in the aligned and noiseless case, the optimal risk is achieved when $\lambda < 0$ (predicted by Theorem 4). The noisy case is presented in Appendix D.

We illustrate the results of Theorem 1 in Figure 2 (noiseless case) and Figure 8 (noisy case) for both discrete and continuous design for $\boldsymbol{d}_x$ and $\boldsymbol{d}_\beta$ with $\boldsymbol{\Sigma}_x = \operatorname{diag}(\boldsymbol{d}_x), \boldsymbol{\Sigma}_\beta = \operatorname{diag}(\boldsymbol{d}_\beta)$ and $\boldsymbol{\Sigma}_w = \boldsymbol{I}$ (see design details in Appendix D). Note that Assumption 1 specifies a joint relation between $\boldsymbol{d}_x(= \boldsymbol{d}_{x/w})$ and $\boldsymbol{d}_\beta(= \boldsymbol{d}_{w\beta})$. In the following section, we mainly consider the three following relations, which allow us to precisely determine the sign of $\lambda_{\text{opt}}$.

**Definition 2.** *For two vectors $\boldsymbol{a}, \boldsymbol{b} \in \mathbb{R}^p$, we say $\boldsymbol{a}$ is aligned (misaligned) with $\boldsymbol{b}$ if the order of $\boldsymbol{a}$ is the same as (reverse of) the order of $\boldsymbol{b}$, i.e., $a_i \geq a_j$ iff $b_i \geq (\leq) b_j$ for all $i, j$. Additionally, we say $\boldsymbol{a}$ and $\boldsymbol{b}$ have random relation if given the order of one vector, the order of the other is uniformly permuted at random.*

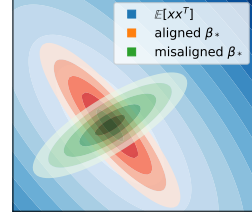

Intuitively, aligned $\boldsymbol{d}_x$ and $\boldsymbol{d}_\beta$ implies that when one component in $\boldsymbol{d}_x$ has large magnitude, then so does the corresponding component in $\boldsymbol{d}_\beta$, and vice versa (see Figure 3). In Figure 2, we plot the prediction risk of all three joint relations defined above (see Appendix D for details).

Figure 3: Alignment between $\boldsymbol{x}$ and $\boldsymbol{\beta}_*$ in 2D.

Theorem 1 allows us to compute the risk of the generalized ridge estimator $\hat{\boldsymbol{\beta}}_\lambda$ and also its ridgeless limit, i.e., the minimum $\|\hat{\boldsymbol{\beta}}\|_{\boldsymbol{\Sigma}_w}$ norm interpolant (taking $\boldsymbol{\Sigma}_w = \boldsymbol{I}$ yields the min $\ell_2$ norm solution).

**Connection to PCR estimator.** Note that the principal component regression (PCR) estimator is closely related to the ridgeless estimator in the following sense: picking the leading $\theta p$ eigenvectors of $\boldsymbol{\Sigma}_x$ (for some $\theta \in [0, 1]$) is equivalent to setting the remaining $(1 - \theta)p$ eigenvalues of $\boldsymbol{\Sigma}_w$ to be infinity [HG83]. The following corollary characterizes the prediction risk of the PCR estimator $\hat{\beta}_\theta$:

**Corollary 3.** *Given Assumption 1 and $\boldsymbol{\Sigma}_w = \boldsymbol{I}$, and $h$ has continuous and strictly increasing quantile function $Q_h$. Then for all $\theta \in (0, 1]$, as $n, p \to \infty$,*

$$
\tilde{\mathbb{E}}\left(\tilde{y} - \tilde{x}\hat{\boldsymbol{\beta}}_\theta\right)^2 \quad \overset{p}{\to} \quad
\begin{cases}
\dfrac{m_\theta'(0)}{m_\theta^2(0)} \cdot \left(\gamma \mathbb{E}\dfrac{gh}{(h_\theta \cdot m_\theta(0) + 1)^2} + \tilde{\sigma}^2\right), & \theta\gamma > 1 \\[2mm]
\left(\gamma \mathbb{E}[gh \cdot \mathbb{I}_{h < Q_h(1-\theta)}] + \tilde{\sigma}^2\right)\dfrac{1}{1 - \theta\gamma}, & \theta\gamma < 1
\end{cases}
\tag{4.4}
$$

*where $h_\theta = h \cdot \mathbb{I}_{h \geq Q_h(1-\theta)}$ and $m_\theta(z)$ satisfies $-z = m_\theta^{-1}(z) - \gamma \mathbb{E}h_\theta \cdot (1 + h_\theta \cdot m_\theta(z))^{-1}$.*

*In addition, if $\mathbb{E}[g|h]$ is a decreasing function of $h$, and $h$ has continuous p.d.f., then the asymptotic prediction risk of $\hat{\boldsymbol{\beta}}_\theta$ is a decreasing function of $\theta$ when $\theta\gamma > 1$.*

Corollary 3 confirms double descent under more general settings of $(\boldsymbol{\Sigma}_x, \boldsymbol{\Sigma}_\beta)$ than [XH19], i.e. the risk exhibits a spike as $\theta\gamma \to 1^-$, and then decreases as we further overparameterize by increasing $\theta$. In Section 6 we compare the PCR estimator $\hat{\boldsymbol{\beta}}_\theta$ with the minimum $\|\hat{\boldsymbol{\beta}}\|_{\boldsymbol{\Sigma}_w}$ norm solution.

**Remark.** *The PCR estimator [XH19] and the ridgeless regression estimator (considered in [HMRT19]) are fundamentally different in the following way: in ridgeless regression, increasing the model size corresponds to changing $\gamma$, which also alters the dimensions of $\boldsymbol{\beta}_*$; in contrast, in PCR, increasing $\theta$ does not change the data generating process (which is a more natural setting).*

*In terms of the risk curve, Figure 9(a) shows that the ridgeless regression estimator can exhibit "multiple descent" as $\gamma$ increases, whereas Corollary 3 and Figure 9(b) demonstrate that in the misaligned case, the PCR risk is monotonically decreasing in the overparameterized regime $\theta\gamma > 1$.*

## 5 Analysis of Optimal $\lambda_{\text{opt}}$

In this section, we focus on the optimal weighted ridge estimator and determine the sign of the optimal regularization parameter $\lambda_{\text{opt}}$. Taking the derivatives of (4.1) yields

$$
R'(\lambda) \cdot \frac{m^3(-\lambda)}{2\gamma(m'(-\lambda))^2} = \underbrace{\left(-\tilde{\sigma}^2 \frac{\mathbb{E}\frac{\zeta^2}{(1+\zeta)^3}}{1 - \gamma\mathbb{E}\frac{\zeta^2}{(1+\zeta)^2}}\right)}_{\text{Part 3}} + \underbrace{\left(\mathbb{E}\frac{gh\zeta}{(1+\zeta)^3} - \frac{\gamma\mathbb{E}\frac{\zeta^2}{(1+\zeta)^3}\mathbb{E}\frac{gh}{(1+\zeta)^2}}{1 - \gamma\mathbb{E}\frac{\zeta^2}{(1+\zeta)^2}}\right)}_{\text{Part 4}}, \tag{5.1}
$$

where $\zeta = h \cdot m(-\lambda)$. For certain special cases, we obtain a closed form solution for $\lambda_{\text{opt}}$ (see details in Appendix B.1) and recover the result from [HMRT19, DW18][6] and beyond:

- When $h \overset{\text{a.s.}}{=} c$ (i.e., isotropic features [HMRT19]), the optimal $\lambda_{\text{opt}}$ is achieved at $c/\xi$.
- When $g \overset{\text{a.s.}}{=} c$ (i.e., isotropic signals [DW18]), the optimal $\lambda_{\text{opt}}$ is achieved at $\tilde{\sigma}^2/c$.
- When $\mathbb{E}[g|h] \overset{\text{a.s.}}{=} \mathbb{E}[g]$ (e.g., random order), the optimal $\lambda_{\text{opt}}$ is achieved at $\tilde{\sigma}^2/\mathbb{E}[g]$.

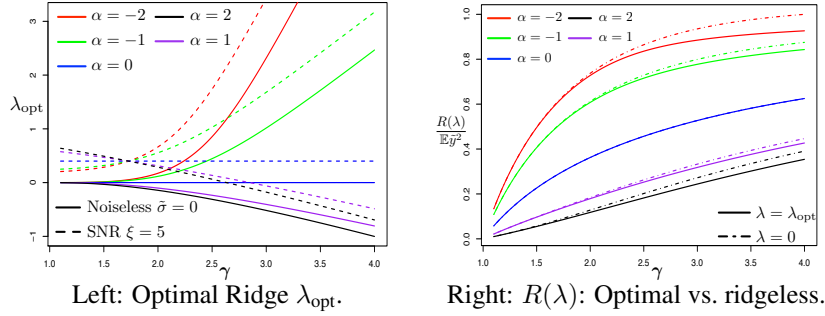

Left: Optimal Ridge $\lambda_{\text{opt}}$.      Right: $R(\lambda)$: Optimal vs. ridgeless.

Figure 4: We set $\boldsymbol{\Sigma}_w = \boldsymbol{I}$ and $\boldsymbol{\Sigma}_\beta = \boldsymbol{\Sigma}_x^\alpha$ where $\boldsymbol{d}_x$ has two point masses on 1 and 5 with probability $3/4$ and $1/4$ respectively. **Left**: optimal $\lambda$; solid lines represents the noiseless case $\tilde{\sigma} = 0$ and dashed lines represents SNR $\xi = 5$. **Right**: prediction risk of the ridgeless ($R(\lambda_{\text{opt}})$, dashed lines) and optimally regularized ($R(\lambda_{\text{opt}})$, solid lines) estimator in the noiseless case. We normalize the prediction risk as $\mathbb{E}\tilde{y}^2 = \mathbb{E}(\tilde{\boldsymbol{x}}^\top \boldsymbol{\beta}_\star)^2$.

Although $\lambda_{\text{opt}}$ may not have a tractable form in general, we may infer the sign of $\lambda_{\text{opt}}$. Note that in (5.1), Part 3 is due to the variance term (Part 1) and Part 4 from the bias term (Part 2) in (3.1). We therefore consider the sign of Part 3 and Part 4 separately in the following theorem.

**Theorem 4.** *Under Assumption 1, we have*

- *Part 3 (derivative of variance) is negative for all $\lambda > -c_0$.*

- *If $\mathbb{E}[g|h]$ is an increasing function of $h$ on its support, then Part 4 (derivative of bias) is positive for all $\lambda > 0$. At $\lambda = 0$, Part 4 is non-negative and achieves $0$ only if $\mathbb{E}[g|h] \overset{a.s.}{=} \mathbb{E}[g]$.*

- *If $\mathbb{E}[g|h]$ is a decreasing function of $h$ on its support, then Part 4 is negative for all $\lambda \in (-c_0, 0)$. At $\lambda = 0$, Part 4 is non-positive and achieves $0$ only if $\mathbb{E}[g|h] \overset{a.s.}{=} \mathbb{E}[g]$.*

The first point in Theorem 4 is consistent with the well-understood variance reduction property of ridge regularization. On the other hand, when the prediction risk is dominated by the bias term (i.e., $\tilde{\sigma}^2 = o(1)$) and both $\boldsymbol{d}_{x/w}$ and $\boldsymbol{d}_{w\beta}$ converge to non-trivial distributions, the second and third point of Theorem 4 reveal the following surprising phenomena (see Figure 2 (a) and (b)):

**M1**   $\lambda_{\text{opt}} < 0$ when $\boldsymbol{d}_{x/w}$ aligns with $\boldsymbol{d}_{w\beta}$, or in general, $\mathbb{E}[g|h]$ is a strictly increasing function of $h$. In the context of standard ridge regression, it means that shrinkage regularization only *increases the bias* in the overparameterized regime when features are informative, i.e., the projection of the signal is large in the directions where the feature variance is large.

**M2**   $\lambda_{\text{opt}} > 0$ when $\boldsymbol{d}_{x/w}$ is misaligned with $\boldsymbol{d}_{w\beta}$, or in general, $\mathbb{E}[g|h]$ is a strictly decreasing function of $h$. This is to say, in standard ridge regression, when features are not informative, i.e., the projection of the signal is small in the directions of large feature variance, shrinkage is beneficial even in the *absence of label noise* (the variance term is zero).

M1 and M2, together with aforementioned special case when $g$ and $h$ have random relation, provide a precise characterization of the sign of $\lambda_{\text{opt}}$. In particular, M1 confirms the "negative ridge" phenomenon empirically observed in [KLS20] and outlines concise conditions under which it occurs. We emphasize that neither M1 nor M2 would be observed when one of $\boldsymbol{\Sigma}_{x/w}$ and $\boldsymbol{\Sigma}_{w\beta}$ is identity. In other words, these observations arise from our more general assumption on $(\boldsymbol{\Sigma}_{x/w}, \boldsymbol{\Sigma}_{w\beta})$.

**Implicit regularization of overparameterization.**   Taking both the bias and variance into account, Theorem 4 suggests a bias-variance tradeoff between Part 3 and Part 4, and $\lambda_{\text{opt}}$ will eventually become positive as $\tilde{\sigma}^2$ increases (i.e., risk is dominated by variance, for which positive $\lambda$ is beneficial). For certain special cases, we can provide a lower bound for the transition from $\lambda_{\text{opt}} < 0$ to $\lambda_{\text{opt}} > 0$.

**Proposition 5.** *Given Assumption 1, let $(h, g) = (1, 1)$ with probability $1 - q$ and $(h, g) = (h_1, g_1)$ with probability $q$, where $h_1 > 1$ and $g_1 > 1$. Denote $\bar{\gamma} = \gamma - 1$. Then $\lambda_{opt} < 0$ if*

$$\tilde{\sigma}^2 < (h_1 - 1)(g_1 - 1)h_1 \cdot \max\left( \frac{(\gamma q - 1)^3 \bar{\gamma}^3 (1 - q)}{(1 - q)\gamma^2(\bar{\gamma}^3 q^2 + (\gamma q - 1)^3 h_1^2)}, \frac{\gamma q (1 - q)\bar{\gamma}^3}{(1 - q)(h_1 + \bar{\gamma})^3 + q h_1^2 \gamma^3} \right).$$

As $q$ approaches 0 or 1, the above upper bound goes to 0 because $\boldsymbol{\Sigma}_x$, $\boldsymbol{\Sigma}_\beta$ becomes closer to $\boldsymbol{I}$. Otherwise, when $\gamma q > 1$, the upper bound suggests $\tilde{\sigma}^2 = O(g_1 \gamma)$ which implies the SNR $\xi = \Omega(h_1/\gamma)$.

Hence, as $\gamma$ increases, $\lambda_{\text{opt}}$ remains negative for a lower SNR, which coincides with the intuition that overparameterization has an implicit effect of $\ell_2$ regularization (Figure 4 Left). Indeed, the following proposition suggests such implicit regularization is only present in the overparameterized regime:

**Proposition 6.** *When $\gamma < 1$, $\lambda_{opt}$ on $(-c_0, \infty)$ is always non-negative under Assumption 1.*

In Figure 4 we confirm our findings in Theorem 4 (for additional results on different distributions see Figure 10). Specifically, we set $\Sigma_w = I, \Sigma_x = \text{diag}(d_x)$ and $\Sigma_\beta = \Sigma_x^\alpha$. As we increase $\alpha$ from negative to positive, the relation between $d_x$ and $d_\beta$ transitions from misaligned to aligned. The left panel shows that the sign of $\lambda_{\text{opt}}$ is the exact opposite to the sign of $\alpha$ in the noiseless case (i.e. the variance is 0), which is consistent with M1 and M2. Moreover, when $d_x$ aligns with $d_\beta$, $\lambda_{\text{opt}}$ decreases as $\gamma$ becomes larger, which agrees with our observation on the implicit $\ell_2$ regularization of overparameterization. Last but not least, in Figure 4 (Right) we see that the optimal ridge regression estimator leads to considerable improvement over the ridgeless estimator. We comment that this improvement becomes more significant as $\gamma$ or condition number of $\Sigma_x$ and $\Sigma_\beta$ increases.

**Risk monotonicity of optimal ridge regression.**   [DS20, Proposition 6] showed that for isotropic data ($\Sigma_x = I$), the asymptotic prediction risk of optimally-tuned ridge regression monotonically increases with $\gamma$. This is to say, under proper regularization, more training data always helps the test performance [KH92]. Here we extend this result to data with general covariance and isotropic $\beta_*$.

**Proposition 7.** *Given $\mathbb{E}[xx^\top] = \Sigma_x$ satisfying Assumption 1 and $\mathbb{E}[\beta_*\beta_*^\top] = \frac{c}{p}I$[7], the asymptotic prediction risk of the optimally-tuned ridge regression estimator (i.e., $\Sigma_w = I$) with $\lambda_{\text{opt}} = \gamma\tilde\sigma^2/c$ is an increasing function of $\gamma \in (0, \infty)$.*

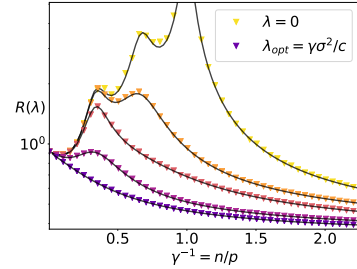

As shown in Figure 5 (where $d_x$ has 3 point masses and $d_\beta = 1$), $\ell_2$ regularization can suppress "multiple descent", and the risk of the optimally-tuned ridge estimator (purple) is monotone w.r.t. $\gamma$. We remark that establishing such characterization under general orientation of $\beta_*$ (anisotropic $\Sigma_\beta$) can be challenging, because the optimal regularization $\lambda_{\text{opt}}$ may not have a convenient closed-form. We leave the analysis for general $\Sigma_\beta$ as future work.

Figure 5: Impact of ridge regularization on the risk curve (SNR=3). Darker color corresponds to larger $\lambda$.

## 6   Optimal Weighting Matrix

Having characterized the optimal regularization strength, we now turn to the optimal weighting matrix $\Sigma_w$. Toward this goal, we additionally require the following assumptions on $(\Sigma_x, \Sigma_\beta, \Sigma_w)$:

**Assumption 2.** *The covariance matrix $\Sigma_x$ and the weighting matrix $\Sigma_w$ share the same set of eigenvectors, i.e., we have the following eigendecompositions: $\Sigma_x = UD_xU^\top$ and $\Sigma_w = UD_wU^\top$, where $U \in \mathbb{R}^{p \times p}$ is orthogonal, and $D_x = \text{diag}(d_x), D_w = \text{diag}(d_w)$.*

Define $\bar{d}_\beta = \text{diag}(U^\top\Sigma_\beta U)$. Note that when $\Sigma_\beta$ also shares the same eigenvector matrix $U$, then $\bar{d}_\beta = d_\beta$, which is simply the eigenvalues of $\Sigma_\beta$.

**Assumption 3.** *Let $d_{x,i}, \bar{d}_{\beta,i}, d_{w,i}$ be the $i$th element of $d_x, \bar{d}_\beta, d_w$ respectively. We assume that the empirical distribution of $(d_{x,i}, \bar{d}_{\beta,i}, d_{w,i})$ jointly converges to $(s, v, s/r)$, where $s, v, r$ are non-negative random variables. Further, there exists constants $c_l, c_u > 0$ independent of $n$ and $p$ such that $\min_i(\min(d_{x,i}, \bar{d}_{\beta,i}, d_{w,i})) \geq c_l$, $\max_i(\max(d_{x,i}, \bar{d}_{\beta,i}, d_{w,i})) \leq c_u$ and $\|\Sigma_\beta\| \leq c_u$.*

For notational convenience, we define $\mathcal{H}_w$ and $\mathcal{H}_r$ to be the sets of all $\Sigma_w$ and $r$, respectively, that satisfy Assumption 2 and Assumption 3. Additionally, let $\mathcal{S}_w$ and $\mathcal{S}_r$ be the subset of $\mathcal{H}_w$ and $\mathcal{H}_r$ such that $r = f(s)$ for some function $f$ (this represents $\Sigma_w \in \mathcal{H}_w$ that only depends on $\Sigma_x$ but not $\Sigma_\beta$). By Assumption 2 and 3, the empirical distribution of $(d_{x/w,i}, d_{w\beta,i})$ jointly converges to $(r, sv/r)$ and satisfies the boundedness requirement in Assumption 1. Thus by Theorem 1 we have:

$$R(r, \lambda) \triangleq \frac{m_r'(-\lambda)}{m_r^2(-\lambda)} \cdot \left(\gamma\mathbb{E}\frac{sv}{(r \cdot m_r(-\lambda) + 1)^2} + \tilde\sigma^2\right), \tag{6.1}$$

where $m_r(-\lambda)$ satisfies the equation $\lambda = m_r^{-1}(-\lambda) - \gamma\mathbb{E}(1 + r \cdot m_r(-\lambda))^{-1}r$. It is clear that when $r \overset{\text{a.s.}}{=} s$, (6.1) reduces to the standard ridge regression with $\Sigma_w = I$, and for $r \overset{\text{a.s.}}{=} 1$, the equation

reduces to the cases of isotropic features ($\Sigma_w = \Sigma_x$). Note that (6.1) indicates that the impact of $\Sigma_\beta$ on the risk is fully captured by $\bar{d}_\beta$. Hence we define $\bar{\Sigma}_\beta = U \operatorname{diag}(\bar{d}_\beta) U^\top$, which corresponds to $r \overset{a.s.}{=} sv$, and is equivalent to $\Sigma_\beta$ when $\Sigma_\beta$ also shares the same eigenvector matrix $U$. In the following subsections, we discuss the optimal $\Sigma_w$ for two types of estimator: the minimum $\|\hat{\beta}\|_{\Sigma_w}$ solution (taking $\lambda \to 0$), and the optimally weighted ridge estimator ($\lambda = \lambda_{\mathrm{opt}}$). Note that the risk for both estimators is scale-invariant over $\Sigma_w$ and $r$. Hence, when we define a specific choice of $(\Sigma_w, r)$, we simultaneously consider all pairs $(c\Sigma_w, {}^r/c)$ for $c > 0$. Finally, we note that the choice of $r \overset{a.s.}{=} s \cdot \mathbb{E}[v|s] \in \mathcal{S}_r$ plays a key role in our analysis, and its corresponding choice of $\Sigma_w$ is given as $\Sigma_w = (f_v(\Sigma_x))^{-1}$, where $f_v(s) \triangleq \mathbb{E}[v|s]$ and $f_v$ applies to the eigenvalues of $\Sigma_x$.

## 6.1 Minimum $\|\hat{\beta}\|_{\Sigma_w}$ solution

Taking the ridgeless limit leads to the following bias-variance decomposition of the prediction risk,

$$\text{Bias:} \quad R_b(r) \triangleq \frac{m_r'(0)}{m_r^2(0)} \cdot \gamma \mathbb{E} \frac{sv}{(r \cdot m_r(0) + 1)^2} \quad \text{Variance:} \quad R_v(r) \triangleq \frac{m_r'(0)}{m_r^2(0)} \cdot \tilde{\sigma}^2.$$

In the previous sections we observe a bias-variance tradeoff in choosing the optimal $\lambda$. Interesting, the following theorem illustrates a similar bias-variance tradeoff in choosing the optimal $\Sigma_w$:

**Theorem 8.** *Given Assumptions 2 and 3,*

- $r \overset{a.s.}{=} sv$ *(i.e., $\Sigma_w = \bar{\Sigma}_\beta^{-1}$) is the optimal choice in $\mathcal{H}_r$ that minimizes the bias function $R_b(r)$. Additionally, $r \overset{a.s.}{=} \mathbb{E}[v|s] \cdot s$ (i.e., $\Sigma_w = (f_v(\Sigma_x))^{-1}$) is the optimal in $\mathcal{S}_r$ that minimizes $R_b(r)$.*

- $r \overset{a.s.}{=} 1$ *(i.e., $\Sigma_w = \Sigma_x$) is optimal in both $\mathcal{S}_r$ and $\mathcal{H}_r$ that minimizes the variance function $R_v(r)$.*

Theorem 8 implies that the variance is minimized when $\Sigma_w = \Sigma_x$. Since the variance term does not depend on $\beta_\star$, it is not surprising that the optimal $\Sigma_w$ is also independent of $\Sigma_\beta$. Furthermore, this result is consistent with the intuition that to minimize the variance, $\hat{\beta}_\lambda$ should be penalized more in the higher variance directions of $\Sigma_x$, and vice versa. On the other hand, Theorem 8 also implies that the bias is minimized when $d_w = 1/\bar{d}_\beta$ which does not depend on $d_x$. While this characterization may not be intuitive, when $\bar{d}_\beta = d_\beta$ (i.e., $\Sigma_\beta$ also shares the same eigenvector matrix $U$), one analogy is that since the quadratic regularization corresponds to the a Gaussian prior $\mathcal{N}(0, \Sigma_w^{-1})$, it is reasonable to match $\Sigma_w^{-1}$ with the covariance of $\beta_\star$, which gives the maximum a posteriori (MAP) estimate. In general, the optimal $\Sigma_w$ admits a bias-variance tradeoff (i.e., the bias and variance are optimal under different $\Sigma_w$) except for the special case of $\Sigma_x \bar{\Sigma}_\beta = I$.

Additionally, the following proposition demonstrates the advantage of the minimum $\|\hat{\beta}\|_{\Sigma_w}$ solution over the PCR estimator in the noiseless case.

**Proposition 9.** *Given Assumption 2 and 3 and $\tilde{\sigma} = 0$, suppose $s$ and $\mathbb{E}[v|s] \cdot s$ both have continuous and strictly increasing quantile functions. Then the minimum $\|\hat{\beta}\|_{\Sigma_w}$ solution outperforms the PCR estimator for all $\theta \in [0, 1)$ when $\Sigma_w = \bar{\Sigma}_\beta^{-1} \in \mathcal{H}_w$, or when $\Sigma_w = (f_v(\Sigma_x))^{-1} \in \mathcal{S}_w$.*

## 6.2 Optimal weighted ridge estimator

Finally, we consider the optimally-tuned weighted shrinkage and discuss the optimal choice of $\Sigma_w$.

**Theorem 10.** *Suppose Assumptions 2 and 3 hold. Then $r \overset{a.s.}{=} sv$ (i.e., $\Sigma_w = \bar{\Sigma}_\beta^{-1}$) is the optimal solution in $\mathcal{H}_r$ that minimizes $\min_\lambda R(r, \lambda)$. Additionally, $r \overset{a.s.}{=} \mathbb{E}[v|s] \cdot s$ (i.e., $\Sigma_w = (f_v(\Sigma_x))^{-1}$) is the optimal solution in $\mathcal{S}_r$ that minimizes $\min_\lambda R(r, \lambda)$.*

In contrast to the ridgeless setting in Theorem 8, the optimal $d_w$ for general $\lambda_{\mathrm{opt}}$ does not depend on the noise level but only on $\bar{d}_\beta$, the strength of the signal in the directions of the eigenvectors of $\Sigma_x$. We conjecture that this is because in the optimally weighted estimator, $\lambda_{\mathrm{opt}}$ is capable of balancing the bias-variance tradeoff; therefore the weighting matrix may not need to adjust to the label noise and can be chosen solely based on the signal $\beta_\star$. Indeed, as previously discussed, $\Sigma_w = \Sigma_\beta^{-1}$ is a preferable choice of prior under the Bayesian perspective when $d_\beta = \bar{d}_\beta$.

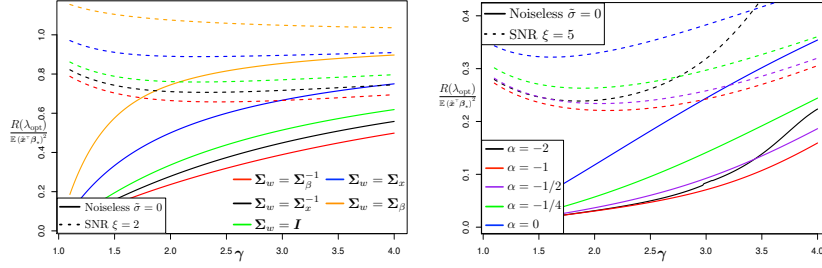

Figure 6: $R(\lambda_{\text{opt}})/\mathbb{E}(\tilde{\boldsymbol{x}}^\top \boldsymbol{\beta}_\star)^2$ against $\gamma$ for various weighting matrix $\boldsymbol{\Sigma}_w$. Solid lines represent the noiseless case $\tilde{\sigma} = 0$ and the dashed lines represent the noisy case with fixed SNR $\xi$. We set $\boldsymbol{d}_x$ to be aligned with $\boldsymbol{d}_\beta$ and **Left**: $\boldsymbol{d}_x$ to have 4 point masses $(1, 2, 3, 4)$ with equal probabilities and $\boldsymbol{d}_\beta$ with 2 point masses on 1 and 5 with probabilities $3/4$ and $1/4$, respectively; **Right**: $\boldsymbol{d}_x$ has 2 point masses on 1 and 5 with probabilities $3/4$ and $1/4$, respectively, and $\boldsymbol{\Sigma}_\beta = \boldsymbol{\Sigma}_x^2$; we set $\boldsymbol{\Sigma}_w = \boldsymbol{\Sigma}_\beta^\alpha$.

Theorem 10 is supported by Figure 6, where we plot the prediction risk of the generalized ridge regression estimator under different $\boldsymbol{\Sigma}_w$ and optimally tuned $\lambda_{\text{opt}}$. We consider a simple discrete construction for aligned $\boldsymbol{d}_x$ and $\boldsymbol{d}_\beta (= \bar{\boldsymbol{d}}_\beta)$. On the left panel, we enumerate a few standard choices of $\boldsymbol{\Sigma}_w$: $\boldsymbol{\Sigma}_x, \boldsymbol{\Sigma}_\beta, \boldsymbol{I}, \boldsymbol{\Sigma}_x^{-1}$ and the optimal choice $\boldsymbol{\Sigma}_\beta^{-1}$ (red). On the right, we take $\boldsymbol{\Sigma}_w$ to be powers of $\boldsymbol{\Sigma}_\beta$ around the optimal $\boldsymbol{\Sigma}_\beta^{-1}$. In both setups, we confirm that $\boldsymbol{\Sigma}_\beta^{-1}$ achieves the lowest risk uniformly over $\gamma$, as predicted by Theorem 10.

Note that our main results require knowledge of $\boldsymbol{\Sigma}_x$ and $\bar{\boldsymbol{\Sigma}}_\beta$. While $\boldsymbol{\Sigma}_x$ can be obtained in a semi-supervised setting using unlabeled data (e.g., [RC15, TCG20]), it is typically difficult to estimate $\bar{\boldsymbol{\Sigma}}_\beta$ directly from data. Without prior knowledge on $\bar{\boldsymbol{\Sigma}}_\beta$, Theorem 10 suggests that $r \overset{\text{a.s.}}{=} \mathbb{E}[v|s] \cdot s$ is the optimal $r$ that only depends on $s$. That is, $\boldsymbol{\Sigma}_w = (f_v(\boldsymbol{\Sigma}_x))^{-1}$ is the optimal $\boldsymbol{\Sigma}_w$ that only depends on $\boldsymbol{\Sigma}_x$. In the special case of $\mathbb{E}[v|s] = \mathbb{E}[v]$, standard ridge regression ($\boldsymbol{\Sigma}_w = \boldsymbol{I}$) is optimal in $\mathcal{S}_w$. When the exact form of $f_v(s)$ is also not known, we may use a polynomial or power function of $s$ to approximate either $f_v(s)$ or $1/f_v(s)$, whose coefficients can be considered as hyper-parameters to be cross-validated. We demonstrate the effectiveness of this heuristic in Figure 7: although our proposed $\boldsymbol{\Sigma}_w = f_v(\boldsymbol{\Sigma}_x)^{-1}$ (blue) is worse than the actual optimal (red) $\boldsymbol{\Sigma}_w = \boldsymbol{\Sigma}_\beta^{-1}$ (same as $\bar{\boldsymbol{\Sigma}}_\beta^{-1}$ due to diagonal design), it is the best choice among weighting matrices that only depend on $\boldsymbol{\Sigma}_x$. In addition, we seek the best approximation of $f_v(s)$ by applying a power transformation on $\boldsymbol{\Sigma}_x$, and we observe that certain powers of $\boldsymbol{\Sigma}_x$ also outperform the standard isotropic regularization.

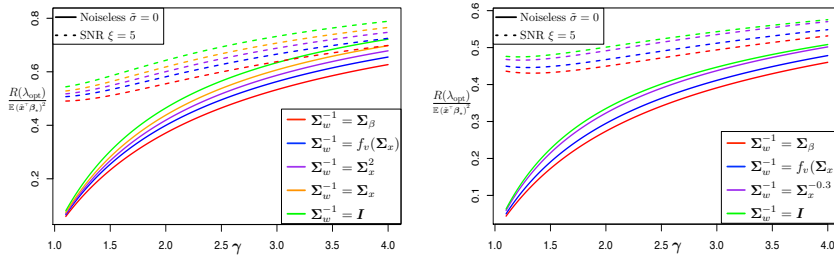

Figure 7: $R(\lambda_{\text{opt}})/\mathbb{E}(\tilde{\boldsymbol{x}}^\top \boldsymbol{\beta}_\star)^2$ against $\gamma$ for various weighting matrix $\boldsymbol{\Sigma}_w$ under noiseless $\tilde{\sigma} = 0$ (solid lines) and noisy setting with fixed SNR $\xi$ (dashed lines). **Left**: We set $f_v(s)$ as an increasing function of $s$ on its support; **Right**: We set $f_v(s)$ as a decreasing function of $s$ on its support. Note that the heuristically chosen weighting matrices often outperform the standard ridge regression estimator (green).

# 7 Conclusion

We provide a precise asymptotic characterization of the prediction risk of generalized ridge regression in the overparameterized regime. Our result greatly generalizes previous high-dimensional analysis of ridge regression, which enables us to discover and theoretically justify various interesting findings, including the negative ridge phenomenon, the implicit regularization of overparameterization, and a concise description of the optimal weighted shrinkage. Future works include extending our analysis to border settings, such as more general eigenvalue conditions [XH19] or the random features regression model [MM19]. Another important direction is to construct weighting matrix $\boldsymbol{\Sigma}_w$ solely from training data that outperforms isotropic shrinkage in the overparameterized regime.

## 8 Broader Impact

This work does not present any foreseeable direct societal consequence.

## Acknowledgement

The authors would like to thank Murat A. Erdogdu, Daniel Hsu and Taiji Suzuki for comments and suggestions, and also anonymous NeurIPS reviewers 1 and 3 for helpful feedback. DW was partially funded by CIFAR, NSERC and LG Electronics. JX was supported by a Cheung-Kong Graduate School of Business Fellowship.

## Footnotes

[2]Some of our results also apply to the underparameterized case, as we explicitly highlight in the sequel.

[3]When $\boldsymbol{\beta}_\star$ is deterministic, $\mathbb{E}_{\tilde{x}, \tilde{\epsilon}, \boldsymbol{\beta}_\star}(\tilde{y} - \tilde{\boldsymbol{x}}^\top\hat{\boldsymbol{\beta}}_\lambda)^2$ reduces to the prediction risk for one fixed $\boldsymbol{\beta}_\star$.

[4]We remark that convergence and uniqueness of AMP and CGMT can be challenging to establish for $\lambda < 0$. Also, to our knowledge the current AMP framework does not handle the *joint* relation between $\boldsymbol{\Sigma}_x$ and $\boldsymbol{\Sigma}_\beta$.

[5]Our negative ridge construction relies on the general notion of *alignment*, which subsumes the spike model.

[6]In [HMRT19], $h \overset{\text{a.s.}}{=} 1$. In [DW18], $\tilde{\sigma}^2 = 1$ and their signal strength $\alpha^2$ is equivalent to $c\gamma$ in our setting.

[7]Note that the parameter scaling differs from the previous setting by $\gamma$ to be consistent with that of [DS20].

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
