[Supplementary Material]

**Table of Contents**

# A  Proofs omitted in Section 4

## A.1  Proof of Theorem 1

We first claim that function $m(-\lambda)$ that satisfies (4.2) is indeed the Stieltjes transform of the limiting distribution of the eigenvalues of $\boldsymbol{X}_{/w}\boldsymbol{X}_{/w}^{\top}$. This is because the empirical distribution of the eigenvalues of $\boldsymbol{\Sigma}_{x/w}$ converges to the distribution of $h$ due to Assumption 1. By the Marchenko-Pastur law, it is straightforward to show that the minimal eigenvalue of $\boldsymbol{X}_{/w}\boldsymbol{X}_{/w}^{\top}$ is lower bounded by $c_0$ as $n \to \infty$. Hence, we have $m(-\lambda) > 0$ for all $\lambda > -c_0$. Then by taking derivatives of (4.2)[7], we know that (4.3) holds. The rest of the proof is to characterize Part 1 and Part 2 in (3.1) and show (4.1).

For Part 1 in (3.1), based on prior works [DW18, XH19], we have

$$\text{Part 1 in (3.1)} \;\xrightarrow{\text{p}}\; \tilde{\sigma}^2 \frac{m'(-\lambda)}{m^2(-\lambda)}. \tag{A.1}$$

Hence, we only need to show that

$$\text{Part 2 in (3.1)} \;\xrightarrow{\text{p}}\; \frac{m'(-\lambda)}{m^2(-\lambda)} \cdot \gamma \mathbb{E}\frac{gh}{(h \cdot m(-\lambda)+1)^2}. \tag{A.2}$$

Towards this goal, we first assume that $\boldsymbol{\Sigma}_{w\beta}$ is invertible and define $\boldsymbol{S} = \boldsymbol{X}_{/w^2\beta}^{\top}\boldsymbol{X}_{/w^2\beta} + \lambda\boldsymbol{\Sigma}_{w\beta}^{-1}$, where $\boldsymbol{X}_{/w^2\beta} = \boldsymbol{X}_{/w}\boldsymbol{\Sigma}_{w\beta}^{-1/2} \sim \mathcal{N}\big(0, \frac{1}{n}\boldsymbol{\Sigma}_{x/w^2\beta}\big)$ and $\boldsymbol{\Sigma}_{x/w^2\beta} = \boldsymbol{\Sigma}_{w\beta}^{-1/2}\boldsymbol{\Sigma}_{x/w}\boldsymbol{\Sigma}_{w\beta}^{-1/2}$. Simplification of Part 2 yields

$$\frac{\lambda^2}{n}\operatorname{tr}\big(\boldsymbol{\Sigma}_{x/w^2\beta}\boldsymbol{S}^{-2}\big).$$

To analyze the above quantity, we adopt the similar strategy used in [LP11] and first characterize a related quantity $\frac{1}{n}\operatorname{tr}\big(\boldsymbol{X}_{/w^2\beta}^{\top}\boldsymbol{X}_{/w\beta^2}\big)$. Note that, on one hand, we know that

$$
\begin{aligned}
\frac{1}{n}\operatorname{tr}\big(\boldsymbol{S}^{-2}\boldsymbol{X}_{/w^2\beta}^{\top}\boldsymbol{X}_{/w^2\beta}\big) &= \frac{1}{n}\operatorname{tr}\big(\boldsymbol{S}^{-1}-\lambda\boldsymbol{S}^{-2}\boldsymbol{\Sigma}_{w\beta}^{-1}\big) \\
&= \frac{1}{n}\operatorname{tr}\big(\boldsymbol{\Sigma}_{w\beta}(\boldsymbol{X}_{/w}^{\top}\boldsymbol{X}_{/w}+\lambda\boldsymbol{I})^{-1}-\lambda\boldsymbol{\Sigma}_{w\beta}(\boldsymbol{X}_{/w}^{\top}\boldsymbol{X}_{/w}+\lambda\boldsymbol{I})^{-2}\big).
\end{aligned}
\tag{A.3}
$$

On the other hand, let $\boldsymbol{x}_{/w^2\beta,i}$ be the $i$th row of $\boldsymbol{X}_{/w^2\beta}$ and $\boldsymbol{S}_{\backslash i} = \boldsymbol{S} - \boldsymbol{x}_{/w^2\beta,i}\boldsymbol{x}_{/w^2\beta,i}^{\top}$, then we have

$$
\begin{aligned}
\frac{1}{n}\operatorname{tr}\big(\boldsymbol{S}^{-2}\boldsymbol{X}_{/w^2\beta}^{\top}\boldsymbol{X}_{/w^2\beta}\big) &= \frac{1}{n}\sum_{i=1}^{n}\boldsymbol{x}_{/w^2\beta,i}^{\top}\boldsymbol{S}^{-2}\boldsymbol{x}_{/w^2\beta,i} \\
&= \frac{1}{n}\sum_{i=1}^{n}\frac{\lambda^2\boldsymbol{x}_{/w^2\beta,i}^{\top}\boldsymbol{S}_{\backslash i}^{-2}\boldsymbol{x}_{/w^2\beta,i}}{\big(\lambda+\lambda\boldsymbol{x}_{/w^2\beta,i}^{\top}\boldsymbol{S}_{\backslash i}^{-1}\boldsymbol{x}_{/w^2\beta,i}\big)^2},
\end{aligned}
\tag{A.4}
$$

where the last equality holds due to the Matrix Inversion Lemma. Note that from Assumption 1, the eigenvalues of $\boldsymbol{\Sigma}_{x/w}$ is lower bounded and upper bounded away from 0 and $\infty$. Also, $\|\boldsymbol{\Sigma}_{w\beta}\|$ is bounded away from $\infty$. Hence, by the Marchenko–Pastur law, we know that

$$\Big\|\lambda^2\boldsymbol{\Sigma}_{x/w}^{1/2}\boldsymbol{\Sigma}_{w\beta}^{-1/2}\boldsymbol{S}_{\backslash i}^{-2}\boldsymbol{\Sigma}_{w\beta}^{-1/2}\boldsymbol{\Sigma}_{x/w}^{1/2}\Big\| = \Big\|\lambda^2\boldsymbol{\Sigma}_{x/w}^{1/2}(\boldsymbol{X}_{/w}^{\top}\boldsymbol{X}_{/w}+\lambda\boldsymbol{I})^{-1}\boldsymbol{\Sigma}_{w\beta}(\boldsymbol{X}_{/w}^{\top}\boldsymbol{X}_{/w}+\lambda\boldsymbol{I})^{-1}\boldsymbol{\Sigma}_{x/w}^{1/2}\Big\|$$

and

$$\Big\|\lambda\boldsymbol{\Sigma}_{x/w}^{1/2}\boldsymbol{\Sigma}_{w\beta}^{-1/2}\boldsymbol{S}_{\backslash i}^{-1}\boldsymbol{\Sigma}_{w\beta}^{-1/2}\boldsymbol{\Sigma}_{x/w}^{1/2}\Big\| = \Big\|\lambda\boldsymbol{\Sigma}_{x/w}^{1/2}(\boldsymbol{X}_{/w}^{\top}\boldsymbol{X}_{/w}+\lambda\boldsymbol{I})^{-1}\boldsymbol{\Sigma}_{x/w}^{1/2}\Big\|$$

are upper bounded away from $\infty$ for any $\lambda > -c_0$[8]. Furthermore, observe that $\boldsymbol{x}_{/w^2\beta,i}$ is independent of $\boldsymbol{S}_{\backslash i}$ and(A.4). Hence by Lemma 2.1 in [LP11], we can show that

$$
\frac{1}{n}\operatorname{tr}\left(\boldsymbol{S}^{-2}\boldsymbol{X}_{/w^2\beta}^{\top}\boldsymbol{X}_{/w^2\beta}\right) \;\xrightarrow{\text{p}}\; \frac{1}{n}\sum_{i=1}^{n}\frac{\frac{\lambda^2}{n}\operatorname{tr}\left(\boldsymbol{\Sigma}_{x/w^2\beta}\boldsymbol{S}_{\backslash i}^{-2}\right)}{\left(\lambda + \frac{\lambda}{n}\operatorname{tr}\left(\boldsymbol{\Sigma}_{x/w^2\beta}\boldsymbol{S}_{\backslash i}^{-1}\right)\right)^2},\;\forall\lambda > -c_0.
$$

Next, we replace $\boldsymbol{S}_{\backslash i}^{-1}$ by $\boldsymbol{S}^{-1}$ and show the difference made by this rank-1 perturbation is negligible. From the Matrix Inversion Lemma, we have

$$
\sup_i\left|\frac{\lambda}{n}\operatorname{tr}\left(\boldsymbol{\Sigma}_{x/w^2\beta}\boldsymbol{S}_{\backslash i}^{-1}\right) - \frac{\lambda}{n}\operatorname{tr}\left(\boldsymbol{\Sigma}_{x/w^2\beta}\boldsymbol{S}^{-1}\right)\right|
$$

$$
= \sup_i \frac{\lambda}{n}\frac{\boldsymbol{x}_{/w^2\beta,i}^{\top}\boldsymbol{S}_{\backslash i}^{-1}\boldsymbol{\Sigma}_{x/w^2\beta}\boldsymbol{S}_{\backslash i}^{-1}\boldsymbol{x}_{/w^2\beta,i}}{1 + \boldsymbol{x}_{/w^2\beta,i}^{\top}\boldsymbol{S}_{\backslash i}^{-1}\boldsymbol{x}_{/w^2\beta,i}}
$$

$$
\leq \sup_i \frac{\lambda}{n}\frac{\|\boldsymbol{x}_{/w^2\beta,i}^{\top}\boldsymbol{S}_{\backslash i}^{-1}\boldsymbol{\Sigma}_{x/w^2\beta}\boldsymbol{S}_{\backslash i}^{-1/2}\|\cdot\|\boldsymbol{S}_{\backslash i}^{-1/2}\boldsymbol{x}_{/w^2\beta,i}\|}{1 + \boldsymbol{x}_{/w^2\beta,i}^{\top}\boldsymbol{S}_{\backslash i}^{-1}\boldsymbol{x}_{/w^2\beta,i}}
$$

$$
\leq \sup_i \frac{\lambda}{n}\frac{\|\boldsymbol{x}_{/w^2\beta,i}^{\top}\boldsymbol{S}_{\backslash i}^{-1/2}\|\cdot\|\boldsymbol{S}_{\backslash i}^{-1/2}\boldsymbol{\Sigma}_{x/w^2\beta}\boldsymbol{S}_{\backslash i}^{-1/2}\|\cdot\|\boldsymbol{S}_{\backslash i}^{-1/2}\boldsymbol{x}_{/w^2\beta,i}\|}{1 + \boldsymbol{x}_{/w^2\beta,i}^{\top}\boldsymbol{S}_{\backslash i}^{-1}\boldsymbol{x}_{/w^2\beta,i}}
$$

$$
\leq \|\boldsymbol{\Sigma}_{x/w}\|\cdot\sup_i\lambda\left\|\left(\boldsymbol{X}_{/w}^{\top}\boldsymbol{X}_{/w} + \lambda\boldsymbol{I} - \boldsymbol{x}_{/w,i}\boldsymbol{x}_{/w,i}^{\top}\right)^{-1}\right\|\cdot\sup_i\frac{1}{n}\frac{\boldsymbol{x}_{/w^2\beta,i}^{\top}\boldsymbol{S}_{\backslash i}^{-1}\boldsymbol{x}_{/w^2\beta,i}}{1 + \boldsymbol{x}_{/w^2\beta,i}^{\top}\boldsymbol{S}_{\backslash i}^{-1}\boldsymbol{x}_{/w^2\beta,i}}
$$

$$
\leq O_{\text{p}}\left(\frac{1}{n}\right).
$$

Similarly,

$$
\sup_i\left|\frac{\lambda^2}{n}\operatorname{tr}\left(\boldsymbol{\Sigma}_{x/w^2\beta}\boldsymbol{S}_{\backslash i}^{-2}\right) - \frac{\lambda^2}{n}\operatorname{tr}\left(\boldsymbol{\Sigma}_{x/w^2\beta}\boldsymbol{S}^{-2}\right)\right|
$$

$$
= \sup_i\left|\frac{\lambda^2}{n}\operatorname{tr}\left(\boldsymbol{\Sigma}_{x/w^2\beta}\boldsymbol{S}_{\backslash i}^{-1}\left(\boldsymbol{S}_{\backslash i}^{-1} - \boldsymbol{S}^{-1}\right)\right)\right| + \sup_i\left|\frac{\lambda^2}{n}\operatorname{tr}\left(\boldsymbol{\Sigma}_{x/w^2\beta}\left(\boldsymbol{S}_{\backslash i}^{-1} - \boldsymbol{S}^{-1}\right)\boldsymbol{S}^{-1}\right)\right|
$$

$$
\leq \|\boldsymbol{\Sigma}_{x/w}\|\cdot\|\boldsymbol{\Sigma}_{w\beta}\|\left(\sup_i\lambda\left\|\left(\boldsymbol{X}_{/w}^{\top}\boldsymbol{X}_{/w} + \lambda\boldsymbol{I} - \boldsymbol{x}_{/w,i}\boldsymbol{x}_{/w,i}^{\top}\right)^{-1}\right\| + \lambda\left\|\left(\boldsymbol{X}_{/w}^{\top}\boldsymbol{X}_{/w} + \lambda\boldsymbol{I}\right)^{-1}\right\|\right)
$$

$$
\times\sup_i\lambda\left\|\left(\boldsymbol{X}_{/w}^{\top}\boldsymbol{X}_{/w} + \lambda\boldsymbol{I} - \boldsymbol{x}_{/w,i}\boldsymbol{x}_{/w,i}^{\top}\right)^{-1}\right\|\cdot\sup_i\frac{1}{n}\frac{\boldsymbol{x}_{/w^2\beta,i}^{\top}\boldsymbol{S}_{\backslash i}^{-1}\boldsymbol{x}_{/w^2\beta,i}}{1 + \boldsymbol{x}_{/w^2\beta,i}^{\top}\boldsymbol{S}_{\backslash i}^{-1}\boldsymbol{x}_{/w^2\beta,i}}
$$

$$
= O_{\text{p}}\left(\frac{1}{n}\right).
$$

Hence, we have

$$
\frac{1}{n}\operatorname{tr}\left(\boldsymbol{S}^{-2}\boldsymbol{X}_{/w^2\beta}^{\top}\boldsymbol{X}_{/w^2\beta}\right) \;\xrightarrow{\text{p}}\; \frac{\frac{\lambda^2}{n}\operatorname{tr}\left(\boldsymbol{\Sigma}_{x/w^2\beta}\boldsymbol{S}^{-2}\right)}{\left(\lambda + \frac{\lambda}{n}\operatorname{tr}\left(\boldsymbol{\Sigma}_{x/w^2\beta}\boldsymbol{S}^{-1}\right)\right)^2}
$$

$$
= \frac{\frac{\lambda^2}{n}\operatorname{tr}\left(\boldsymbol{\Sigma}_{x/w^2\beta}\boldsymbol{S}^{-2}\right)}{\left(\lambda + \frac{\lambda}{n}\operatorname{tr}\left(\boldsymbol{D}_{x/w}\left(\boldsymbol{X}_{/w}^{\top}\boldsymbol{X}_{/w} + \lambda\boldsymbol{I}\right)^{-1}\right)\right)^2}
$$

$$
\xrightarrow{\text{p}}\; \frac{\frac{\lambda^2}{n}\operatorname{tr}\left(\boldsymbol{\Sigma}_{x/w^2\beta}\boldsymbol{S}^{-2}\right)}{\left(\frac{1}{m(-\lambda)}\right)^2},\quad \forall\lambda > -c_0, \tag{A.5}
$$

where the last equality used the following known results in [LP11, DW18, XH19]:

$$
\frac{\lambda}{n}\operatorname{tr}\left(\boldsymbol{\Sigma}_{x/w}\left(\boldsymbol{X}_{/w}^{\top}\boldsymbol{X}_{/w} + \lambda\boldsymbol{I}\right)^{-1}\right) \;\xrightarrow{\text{p}}\; \frac{1}{m(-\lambda)} - \lambda.
$$

Combine (A.3) and (A.5), we have

$$\text{Part 2} \quad \overset{\text{p}}{\to} \quad \frac{1}{m^2(-\lambda)} \cdot \frac{1}{n} \operatorname{tr}\left(\boldsymbol{\Sigma}_{w\beta}(\boldsymbol{X}_{/w}^\top \boldsymbol{X}_{/w} + \lambda \boldsymbol{I})^{-1} - \lambda \boldsymbol{\Sigma}_{w\beta}(\boldsymbol{X}_{/w}^\top \boldsymbol{X}_{/w} + \lambda \boldsymbol{I})^{-2}\right) \quad \forall \lambda > -c_0.$$

Our next step is to characterize $\frac{1}{n} \operatorname{tr}\left(\boldsymbol{\Sigma}_{w\beta}(\boldsymbol{X}_{/w}^\top \boldsymbol{X}_{/w} + \lambda \boldsymbol{I})^{-1}\right)$. From Theorem 1 in [RM11], for any deterministic sequence of matrices $\boldsymbol{\Theta}_n$ such that $\frac{1}{n} \operatorname{tr}\left((\boldsymbol{\Theta}_n^\top \boldsymbol{\Theta}_n)^{1/2}\right)$ is finite, we know that as $n, p \to \infty$,

$$\frac{1}{n} \operatorname{tr}\left(\boldsymbol{\Theta}_n \left(\boldsymbol{X}_{/w}^\top \boldsymbol{X}_{/w} - z\boldsymbol{I}\right)^{-1}\right) \overset{\text{a.s.}}{\to} \frac{1}{n} \operatorname{tr}\left(\boldsymbol{\Theta}_n \left(c_n(z)\boldsymbol{\Sigma}_{x/w} - z\boldsymbol{I}\right)^{-1}\right), \quad \forall z \in \mathbb{C}^+ - \mathbb{R}^+,$$

where $c_n(z)$ satisfies

$$c_n(z) = 1 - \gamma \mathbb{E}\frac{h_n c_n(z)}{h_n c_n(z) - z},$$

and $h_n$ follows the empirical distribution of $\boldsymbol{D}_{x/w}$. Hence, it is clear that $c_n(z) \to -zm(z)$ for all $z \in \mathbb{C}^+ - \mathbb{R}^+$ due to (4.2) and the dominated convergence theorem. Now let $\boldsymbol{\Theta}_n = \boldsymbol{\Sigma}_{w\beta}$. Since $\boldsymbol{\Sigma}_{w\beta}$ is a positive semi-definite matrix, we have

$$\frac{1}{n} \operatorname{tr}\left((\boldsymbol{\Sigma}_{w\beta}^\top \boldsymbol{\Sigma}_{w\beta})^{1/2}\right) = \frac{1}{n} \operatorname{tr}(\boldsymbol{\Sigma}_{w\beta}) \leq \frac{d}{n} c_u < \infty.$$

Therefore, applying Theorem 1 in [RM11] yields

$$\begin{aligned}
\frac{1}{n} \operatorname{tr}\left(\boldsymbol{\Sigma}_{w\beta}\left(\boldsymbol{X}_{/w}^\top \boldsymbol{X}_{/w} - z\boldsymbol{I}\right)^{-1}\right) \quad &\overset{\text{a.s.}}{\to} \quad \frac{1}{n} \operatorname{tr}\left(\boldsymbol{\Sigma}_{w\beta}\left(-zm(z) \cdot \boldsymbol{\Sigma}_{x/w} - z\boldsymbol{I}\right)^{-1}\right) \\
&= \quad \frac{1}{n} \operatorname{tr}\left(\boldsymbol{U}_{x/w}\boldsymbol{\Sigma}_{w\beta}\boldsymbol{U}_{x/w}^\top\left(-zm(z) \cdot \boldsymbol{D}_{x/w} - z\boldsymbol{I}\right)^{-1}\right) \\
&= \quad \frac{1}{-zn} \sum_{i=1}^d \frac{d_{w\beta,i}}{d_{x/w,i}m(z) + 1}, \quad \forall z \in \mathbb{C}^+ - \mathbb{R}^+.
\end{aligned}$$

From Assumption 1 and dominated convergence theorem, we have

$$\frac{\lambda}{n} \operatorname{tr}\left(\boldsymbol{\Sigma}_{w\beta}\left(\boldsymbol{X}_{/w}^\top \boldsymbol{X}_{/w} + \lambda \boldsymbol{I}\right)^{-1}\right) \overset{\text{a.s.}}{\to} \gamma \mathbb{E}\frac{g}{h \cdot m(-\lambda) + 1}, \quad \forall -\lambda \in \mathbb{C}^+ - \mathbb{R}^+. \tag{A.6}$$

Note that both $\boldsymbol{\Sigma}_{w\beta}$ and $\left(\boldsymbol{X}_{/w}^\top \boldsymbol{X}_{/w} + \lambda \boldsymbol{I}\right)^{-1}$ are positive semi-definite matrices, and thus

$$\begin{aligned}
\frac{\lambda}{n} \operatorname{tr}\left(\boldsymbol{\Sigma}_{w\beta}\left(\boldsymbol{X}_{/w}^\top \boldsymbol{X}_{/w} + \lambda \boldsymbol{I}\right)^{-1}\right) \quad &\leq \quad \lambda \left\|\left(\boldsymbol{X}_{/w}^\top \boldsymbol{X}_{/w} + \lambda \boldsymbol{I}\right)^{-1}\right\| \cdot \frac{1}{n} \operatorname{tr}(\boldsymbol{\Sigma}_{w\beta}) \\
&\leq \quad \lambda \left\|\left(\boldsymbol{X}_{/w}^\top \boldsymbol{X}_{/w} + \lambda \boldsymbol{I}\right)^{-1}\right\| \cdot \frac{d}{n} c_u.
\end{aligned}$$

Hence $\frac{\lambda}{n} \operatorname{tr}\left(\boldsymbol{\Sigma}_{w\beta}\left(\boldsymbol{X}_{/w}^\top \boldsymbol{X}_{/w} + \lambda \boldsymbol{I}\right)^{-1}\right)$ is bounded on $\lambda > -c_0$; by the dominated convergence theorem, we can extend (A.6) to $\lambda > -c_0$ and conclude that

$$\frac{\lambda}{n} \operatorname{tr}\left(\boldsymbol{\Sigma}_{w\beta}(\boldsymbol{X}_{/w}^\top \boldsymbol{X}_{/w} + \lambda \boldsymbol{I})^{-1}\right) \overset{\text{a.s.}}{\to} \gamma \mathbb{E}\frac{g}{h \cdot m(-\lambda) + 1}, \quad \forall \lambda > -c_0.$$

It is straightforward to check $\frac{1}{n} \operatorname{tr}\left(\boldsymbol{\Sigma}_{w\beta}(\boldsymbol{X}_{/w}^\top \boldsymbol{X}_{/w} + \lambda \boldsymbol{I})^{-2}\right)$ is bounded as well. With arguments similar to [DW18] and [HMRT19], we have

$$\frac{1}{n} \operatorname{tr}\left(\boldsymbol{\Sigma}_{w\beta}(\boldsymbol{X}_{/w}^\top \boldsymbol{X}_{/w} + \lambda \boldsymbol{I})^{-2}\right) = -\frac{\partial \frac{1}{n} \operatorname{tr}\left(\boldsymbol{\Sigma}_{w\beta}(\boldsymbol{X}_{/w}^\top \boldsymbol{X}_{/w} + \lambda \boldsymbol{I})^{-1}\right)}{\partial \lambda}.$$

We therefore arrive at the desired result

$$\frac{\lambda}{n} \operatorname{tr}\left(\boldsymbol{\Sigma}_{w\beta}(\boldsymbol{X}_{/w}^\top \boldsymbol{X}_{/w} + \lambda \boldsymbol{I})^{-2}\right) \overset{\text{a.s.}}{\to} \frac{\gamma}{\lambda} \mathbb{E}\frac{g}{h \cdot m(-\lambda) + 1} - \gamma \mathbb{E}\frac{g \cdot m'(-\lambda)}{(h \cdot m(-\lambda) + 1)^2}.$$

Combining the above calculations, we know (A.2) holds when $\Sigma_{w\beta}$ is invertible. Finally, we extend (A.2) to the case when $\Sigma_{w\beta}$ is not invertible. For any $\epsilon > 0$, we let $\Sigma_{w\beta}^{\epsilon} = \Sigma_{w\beta} + \epsilon I$. Then, from the above analysis, we have

$$\frac{\lambda^2}{n} \operatorname{tr}\left( \Sigma_{x/w} \left( X_{/w}^{\top} X_{/w} + \lambda I \right)^{-1} \Sigma_{w\beta}^{\epsilon} \left( X_{/w}^{\top} X_{/w} + \lambda I \right)^{-1} \right) \overset{\text{a.s.}}{\to} \frac{m'(-\lambda)}{m^2(-\lambda)} \cdot \gamma \mathbb{E} \frac{(g+\epsilon)h}{(h \cdot m(-\lambda) + 1)^2}.$$

Note that the LHS of above equation is decreasing as $\epsilon$ decreases to $0$ and the RHS of above equation is always bounded for any $\epsilon < 1$. Hence by the dominated convergence theorem, we know that (A.2) holds for non-invertible $\Sigma_{w\beta}$ as well.

## A.2 Proof of Corollary 3

We only provide the proof for the overparameterized regime when $\theta\gamma > 1$, because the calculation is straightforward when $\theta\gamma < 1$ (see [XH19]). Since $h$ has continuous strictly increasing quantile function $Q_h$, we know that the $1 - \theta$ quantile of $d_{x/w}$ (which is the threshold of top $\theta p$ elements of $d_{x/w}$) converges to $Q_h(1 - \theta)$. Therefore, the empirical distribution of the top $\theta p$ elements of $d_{x/w}$ and the corresponding $d_{w\beta}$ jointly converges to the conditional distribution of $(h, g)$ given $h \geq Q_h(1 - \theta)$. Hence, we can apply Theorem 1 and obtain that

$$\tilde{\mathbb{E}}\left( \tilde{y} - \tilde{x}\hat{\beta}_\theta \right)^2 \overset{\text{p}}{\to} \frac{m'_\theta(0)}{m_\theta^2(0)} \cdot \left( \gamma\theta\mathbb{E}\left[ \frac{gh}{(h \cdot m_\theta(0) + 1)^2} \Big| h \geq Q_h(1 - \theta) \right] + \gamma\mathbb{E}gh\mathbb{I}_{h < Q_h(1-\theta)} + \tilde{\sigma}^2 \right).$$

Here the extra term $\gamma\mathbb{E}gh\mathbb{I}_{h < Q_h(1-\theta)}$ comes from the "misspecification" by dropping the small $(1 - \theta)p$ number of eigenvalues, and $m_\theta(z)$ should satisfy that

$$-z = \frac{1}{m_\theta(z)} - \gamma\theta\mathbb{E}\left[ \frac{h}{1 + h \cdot m_\theta(z)} \Big| h \geq Q_h(1 - \theta) \right].$$

By replacing the conditional expectation with the normal expectation, we complete the calculation of the asymptotic prediction risk in Corollary 3.

Next, when $\mathbb{E}[g|h]$ is a decreasing function of $h$ and $h$ has continuous p.d.f. denoted by $f(h)$ (in this proof), we show that the asymptotic prediction risk $\frac{m'_\theta(0)}{m_\theta^2(0)} \cdot \left( \gamma\mathbb{E}\left[ \frac{gh}{(h_\theta \cdot m_\theta(0)+1)^2} \right] + \tilde{\sigma}^2 \right) \triangleq R_\theta$ is a decreasing function of $\theta$. Let $q_\theta$ and $m_\theta$ be the shorthand for $Q_h(1 - \theta)$ and $m_\theta(0)$ respectively. Because $Q_h$ is a strictly increasing continuous function and $h$ has continuous p.d.f., we know that $\frac{\partial q_\theta}{\partial \theta}$ exists and is negative. Hence, by the chain rule, we only need to show that $\frac{\partial R_\theta}{\partial q_\theta} > 0$, which is equivalent to

$$0 < \left( -\frac{\mathbb{E}[g|h]hf(h)}{(hm_\theta + 1)^2}\Big|_{h=q_\theta} + \mathbb{E}[g|h]hf(h)\Big|_{h=q_\theta} - 2\mathbb{E}\frac{gh_\theta^2}{(h_\theta m_\theta + 1)^3} \cdot \frac{\partial m_\theta}{\partial q_\theta} \right) \mathbb{E}\frac{h_\theta m_\theta}{(h_\theta m_\theta + 1)^2}$$

$$- \left( \mathbb{E}\frac{gh}{(h_\theta m_\theta + 1)^2} + \tilde{\sigma}^2 \right) \left( \frac{h^2 m_\theta^2 f(h)}{(hm_\theta + 1)^2}\Big|_{h=q_\theta} - 2\mathbb{E}\frac{h_\theta^2 m_\theta}{(h_\theta m_\theta + 1)^3} \cdot \frac{\partial m_\theta}{\partial q_\theta} \right), \qquad \text{(A.7)}$$

where we use the fact that

$$\frac{m'_\theta(0)}{m_\theta^2(0)} = \left( 1 - \gamma\mathbb{E}\left( \frac{h_\theta m_\theta}{h_\theta m_\theta + 1} \right)^2 \right)^{-1} \quad \text{and} \quad 1 = \gamma\mathbb{E}\frac{h_\theta m_\theta}{1 + h_\theta m_\theta}.$$

We simplify the RHS of (A.7) by breaking it into three parts:

$$\text{RHS of (A.7)} = \underbrace{\left( \mathbb{E}[g|h]f(h)\frac{h^2 m_\theta^2(2 + hm_\theta)}{(hm_\theta + 1)^2}\Big|_{h=q_\theta} \right) \mathbb{E}\frac{h_\theta}{(h_\theta m_\theta + 1)^2} - \left( \frac{h^2 m_\theta^2 f(h)}{(hm_\theta + 1)^2}\Big|_{h=q_\theta} \right) \mathbb{E}\frac{\mathbb{E}[g|h]h_\theta}{(h_\theta m_\theta + 1)^2}}_{\text{part (i)}}$$

$$+ \underbrace{\frac{2}{m_\theta^2}\frac{\partial m_\theta}{\partial q_\theta} \cdot \left( \mathbb{E}\frac{h_\theta^2 m_\theta^2}{(h_\theta m_\theta + 1)^3}\mathbb{E}\frac{gh_\theta m_\theta}{(h_\theta m_\theta + 1)^2} - \mathbb{E}\frac{gh_\theta^2 m_\theta^2}{(h_\theta m_\theta + 1)^3}\mathbb{E}\frac{h_\theta m_\theta}{(h_\theta m_\theta + 1)^2} \right)}_{\text{part (ii)}}$$

$$+ \underbrace{\left( \mathbb{E}gh\mathbb{I}_{h < q_\theta} + \tilde{\sigma}^2 \right) \left( 2\mathbb{E}\frac{h_\theta^2 m_\theta}{(h_\theta m_\theta + 1)^3} \cdot \frac{\partial m_\theta}{\partial q_\theta} - \frac{h^2 m_\theta^2 f(h)}{(hm_\theta + 1)^2}\Big|_{h=q_\theta} \right)}_{\text{part (iii)}} \qquad \text{(A.8)}$$

To show part (i) is positive, note that since $\mathbb{E}[g|h]$ is a decreasing function of $h$, we have

$$\mathbb{E}\frac{\mathbb{E}[g|h]h_\theta}{(h_\theta m_\theta + 1)^2} \leq \mathbb{E}[g|h]\Big|_{h=q_\theta} \cdot \mathbb{E}\frac{h_\theta}{(h_\theta m_\theta + 1)^2},$$

Therefore,

$$\text{part (i)} \geq \mathbb{E}[g|h]\Big|_{h=q_\theta} \cdot \mathbb{E}\frac{h_\theta}{(h_\theta m_\theta + 1)^2} \cdot \left(\frac{h^2 m_\theta^2 f(h)}{(hm_\theta + 1)}\Big|_{h=q_\theta}\right)$$

Hence part (i) is positive because $q_\theta < \sup h$.

To show that part (ii) is non-negative, observe that by taking derivatives with respect to $q_\theta$ on both sides of $1 = \gamma\mathbb{E}\frac{h_\theta m_\theta}{h_\theta m_\theta + 1}$, we have

$$\mathbb{E}\frac{h_\theta}{(h_\theta m_\theta + 1)^2} \cdot \frac{\partial m_\theta}{\partial q_\theta} = \frac{hm_\theta f(h)}{hm_\theta + 1}\Big|_{h=q_\theta}. \tag{A.9}$$

Hence, we know that $\frac{\partial m_\theta}{\partial q_\theta} > 0$. What remains is to show that

$$\mathbb{E}\frac{h_\theta^2 m_\theta^3}{(h_\theta m_\theta + 1)^2}\mathbb{E}\frac{\mathbb{E}[g|h]h_\theta m_\theta}{(h_\theta m_\theta + 1)^2} \geq \mathbb{E}\frac{\mathbb{E}[g|h]h_\theta^2 m_\theta^2}{(h_\theta m_\theta + 1)^3}\mathbb{E}\frac{h_\theta m_\theta}{(h_\theta m_\theta + 1)^2}. \tag{A.10}$$

Denote the probability measure of $h$ as $\mu$ and let $\tilde\mu$ be the new measure of $h_\theta m_\theta \mu \cdot [(h_\theta m_\theta + 1)^2\mathbb{E}\frac{h_\theta m_\theta}{(h_\theta m_\theta + 1)^2}]^{-1}$. Let $\tilde h$ be a random variable following the new measure $\tilde\mu$ and $\tilde h_\theta = \tilde h\mathbb{I}_{\tilde h \geq q_\theta}$. Then since $\frac{\tilde h_\theta m_\theta}{\tilde h_\theta m_\theta + 1}$ is an increasing function of $\tilde h$ and $\mathbb{E}[g|h = \tilde h]$ is a decreasing function of $\tilde h$, we have

$$\mathbb{E}\frac{\tilde h_\theta m_\theta}{\tilde h_\theta m_\theta + 1} \cdot \mathbb{E}\Big(\mathbb{E}[g|h = \tilde h]\Big) \geq \mathbb{E}\frac{\tilde h_\theta m_\theta \mathbb{E}[g|h = \tilde h]}{\tilde h_\theta m_\theta + 1}.$$

We then change $\tilde h$ back to $h$ and obtain that (A.10) holds. We therefore conclude that part (ii) is non-negative.

To show that part (iii) is non-negative, we only need to confirm that

$$2\mathbb{E}\frac{h_\theta^2 m_\theta}{(h_\theta m_\theta + 1)^3} \cdot \frac{\partial m_\theta}{\partial q_\theta} \geq \frac{h^2 m_\theta^2 f(h)}{(hm_\theta + 1)^2}\Big|_{h=q_\theta}$$

From (A.9), this is equivalent to

$$2\mathbb{E}\frac{h_\theta^2 m_\theta}{(h_\theta m_\theta + 1)^3} \cdot \frac{hm_\theta f(h)}{hm_\theta + 1}\Big|_{h=q_\theta} \geq \frac{h^2 m_\theta^2 f(h)}{(hm_\theta + 1)^2}\Big|_{h=q_\theta} \cdot \mathbb{E}\frac{h_\theta}{(h_\theta m_\theta + 1)^2},$$

which is then equivalent to

$$2\mathbb{E}\frac{h_\theta^2}{(h_\theta m_\theta + 1)^3} \geq \frac{h}{(hm_\theta + 1)}\Big|_{h=q_\theta} \cdot \mathbb{E}\frac{h_\theta}{(h_\theta m_\theta + 1)^2}.$$

The above equation clearly holds because $\frac{h}{hm_\theta + 1}$ is an increasing function of $h$.

The proof of Corollary 3 is completed by combining the above calculations.

# B    Proofs omitted in Section 5

## B.1    Optimal $\lambda_{\text{opt}}$ for simple cases

When $h \overset{\text{a.s.}}{=} c$, then $\zeta = h \cdot m(-\lambda)$ is a single point mass at $c \cdot m(-\lambda)$. Thus (5.1) achieves 0 is equivalent to

$$\mathbb{E}[g] \cdot \left(1 - \gamma\frac{\zeta^2}{(1+\zeta)^2}\right) - \gamma\frac{\zeta}{(1+\zeta)^2}\mathbb{E}[g] - \tilde\sigma^2 m(-\lambda) = 0.$$

which is also equivalent to

$$\frac{\mathbb{E}[g]}{\tilde\sigma^2}\left(1 - \gamma\frac{\zeta}{(1+\zeta)}\right) = m(-\lambda). \tag{B.1}$$

Note that (4.2) is now simplified to

$$1 = \lambda m(-\lambda) + \gamma \frac{\zeta}{1 + \zeta}.$$

Furthermore, under Assumption 1, the SNR can be simplified to

$$\xi = \frac{c\mathbb{E}[g]}{\tilde{\sigma}^2}.$$

Plug the above calculations into (B.1), we have

$$\lambda_{\text{opt}} = \frac{\tilde{\sigma}^2}{\mathbb{E}[g]} = \frac{c}{\xi}.$$

On the other hand, when $g \overset{\text{a.s.}}{=} c$, then (5.1) achieves 0 is equivalent to

$$c\mathbb{E}\frac{\zeta^2}{(1+\zeta)^3} \cdot \left(1 - \gamma\mathbb{E}\frac{\zeta^2}{(1+\zeta)^2}\right) - c\gamma\mathbb{E}\frac{\zeta^2}{(1+\zeta)^3}\mathbb{E}\frac{\zeta}{(1+\zeta)^2} - \tilde{\sigma}^2 m(-\lambda)\mathbb{E}\frac{\zeta^2}{(1+\zeta)^3} = 0,$$

which is equivalent to

$$c\left(1 - \gamma\mathbb{E}\frac{\zeta}{1+\zeta}\right) - \tilde{\sigma}^2 m(-\lambda) = 0.$$

Plug (4.2) in above equation, we recover

$$\lambda_{\text{opt}} = \frac{\tilde{\sigma}^2}{c}.$$

Finally when $\mathbb{E}[g|h] \overset{\text{a.s.}}{=} \mathbb{E}g$, then (5.1) achieving 0 is equivalent to

$$\mathbb{E}g \cdot \mathbb{E}\frac{\zeta^2}{(1+\zeta)^3} \cdot \left(1 - \gamma\mathbb{E}\frac{\zeta^2}{(1+\zeta)^2}\right) - \mathbb{E}g \cdot \gamma\mathbb{E}\frac{\zeta^2}{(1+\zeta)^3}\mathbb{E}\frac{\zeta}{(1+\zeta)^2} - \tilde{\sigma}^2 m(-\lambda)\mathbb{E}\frac{\zeta^2}{(1+\zeta)^3} = 0,$$

which is equivalent to

$$\mathbb{E}g \cdot \left(1 - \gamma\mathbb{E}\frac{\zeta}{1+\zeta}\right) - \tilde{\sigma}^2 m(-\lambda) = 0.$$

Plug (4.2) in above equation yields the desired result

$$\lambda_{\text{opt}} = \frac{\tilde{\sigma}^2}{\mathbb{E}[g]}.$$

## B.2 Proof of Theorem 4

Let $\zeta = h \cdot m(-\lambda)$. Taking derivatives of (4.3) with respect to $\lambda$ on both sides, we have

$$m''(-\lambda) = 2\frac{1 - \gamma\mathbb{E}\frac{\zeta^3}{(1+\zeta)^3}}{1 - \gamma\mathbb{E}\frac{\zeta^2}{(1+\zeta)^2}} \cdot \frac{(m'(-\lambda))^2}{m(-\lambda)}. \tag{B.2}$$

Also, rearranging (4.2) and (4.3) yields

$$\lambda m(-\lambda) = 1 - \gamma\mathbb{E}\frac{\zeta}{1+\zeta}, \tag{B.3}$$

$$m'(-\lambda) = \left(1 - \gamma\mathbb{E}\frac{\zeta^2}{(\zeta+1)^2}\right)^{-1} m^2(-\lambda). \tag{B.4}$$

By (B.2)-(B.4), we have

$$
\begin{aligned}
\frac{\mathrm{d}\, \frac{m'(-\lambda)}{m^2(-\lambda)}}{\mathrm{d}\,\lambda} &= \frac{-m''(-\lambda)m(-\lambda) + 2(m'(-\lambda))^2}{m^3(-\lambda)} \\
&= -\frac{2\gamma(m'(-\lambda))^2}{m^3(-\lambda)} \cdot \frac{1}{1 - \gamma\mathbb{E}\frac{\zeta^2}{(1+\zeta)^2}} \cdot \mathbb{E}\frac{\zeta^2}{(1+\zeta)^3}.
\end{aligned}
\tag{B.5}
$$

Hence with (B.5), it is straightforward to obtain (5.1). In addition, note that $m(-\lambda), m'(-\lambda) > 0$ for all $\lambda > -c_0$. Therefore, from (B.4), we know that $1 - \gamma \mathbb{E}\frac{\zeta^2}{(1+\zeta)^2} > 0$ and thus Part 3 is always negative for all $\lambda > -c_0$.

Next, we analyze the sign of Part 4. Note that

$$\text{Part 4} \gtreqless 0 \quad \Leftrightarrow \quad \left(\frac{1}{\gamma} - \mathbb{E}\frac{\zeta^2}{(1+\zeta)^2}\right)\mathbb{E}\frac{gh\zeta}{(\zeta+1)^3} \gtreqless \mathbb{E}\frac{\zeta^2}{(1+\zeta)^3}\mathbb{E}\frac{gh}{(1+\zeta)^2}$$

$$\Leftrightarrow \quad \left(\frac{\lambda m(-\lambda)}{\gamma} + \mathbb{E}\frac{\zeta}{(1+\zeta)^2}\right)\mathbb{E}\frac{g\zeta^2}{(\zeta+1)^3} \gtreqless \mathbb{E}\frac{\zeta^2}{(1+\zeta)^3}\mathbb{E}\frac{g\zeta}{(1+\zeta)^2},$$

(B.6)

where the last equivalence holds due to (B.3) and $m(-\lambda) > 0$ for all $\lambda > -c_0$. Denote the probability measure of $h$ as $\mu(h)$. We introduce a new probability measure $\tilde{\mu}(h) = \frac{\zeta\mu(h)}{(1+\zeta)^2\mathbb{E}\frac{\zeta}{(1+\zeta)^2}}$. Let $\tilde{h}$ follow this new measure $\tilde{\mu}$ and $\tilde{\zeta} = \tilde{h} \cdot m(-\lambda)$. In addition define $f(h) = \mathbb{E}[g|h]$.

• When $\mathbb{E}[g|h] = f(h)$ is an increasing function of $h$, then for any fixed $m(-\lambda) > 0$, we have

$$\mathbb{E}\frac{f(\tilde{h})\tilde{\zeta}}{1+\tilde{\zeta}} \geq \mathbb{E}\frac{\tilde{\zeta}}{1+\tilde{\zeta}}\mathbb{E}f(\tilde{h}),$$

because both $\frac{\tilde{\zeta}}{1+\tilde{\zeta}}$ and $f(\tilde{h})$ are increasing function of $\tilde{h}$. Then we change $\tilde{h}$ back to $h$ and obtain

$$\mathbb{E}\frac{\mathbb{E}[g|h]\zeta^2}{(\zeta+1)^3}\mathbb{E}\frac{\zeta}{(\zeta+1)^2} \geq \mathbb{E}\frac{\zeta^2}{(1+\zeta)^3}\mathbb{E}\frac{\mathbb{E}[g|h]\zeta}{(\zeta+1)^2}.$$

Hence, for all $\lambda > 0$, we know that Part 4 is positive; at $\lambda = 0$, Part 4 is non-negative. Moreover, the equality in above equation is only achieved when $\mathbb{E}[g|h]$ is constant almost surely or $h$ is constant almost surely, which is equivalent to $\mathbb{E}[g|h] \overset{\text{a.s.}}{=} \mathbb{E}[g]$. Hence, Part 4 is 0 at $\lambda = 0$ only when $\mathbb{E}[g|h] \overset{\text{a.s.}}{=} \mathbb{E}[g]$.

• When $\mathbb{E}[g|h] = f(h)$ is a decreasing function of $h$, then for any fixed $m(-\lambda) > 0$, we have

$$\mathbb{E}\frac{f(\tilde{h})\tilde{\zeta}}{1+\tilde{\zeta}} \leq \mathbb{E}\frac{\tilde{\zeta}}{1+\tilde{\zeta}}\mathbb{E}f(\tilde{h}),$$

due to the fact that $\frac{\tilde{\zeta}}{1+\tilde{\zeta}}$ and $f(\tilde{h})$ have different monotonicity w.r.t. $\tilde{h}$. Replacing $\tilde{h}$ with $h$, we arrive at

$$\mathbb{E}\frac{\mathbb{E}[g|h]\zeta^2}{(\zeta+1)^3}\mathbb{E}\frac{\zeta}{(\zeta+1)^2} \leq \mathbb{E}\frac{\zeta^2}{(1+\zeta)^3}\mathbb{E}\frac{\mathbb{E}[g|h]\zeta}{(\zeta+1)^2}.$$

Hence, for all $\lambda < 0$, we know that Part 4 is negative; at $\lambda = 0$, Part 4 is non-positive. Similarly, Part 4 is 0 at $\lambda = 0$ only when $\mathbb{E}[g|h] \overset{\text{a.s.}}{=} \mathbb{E}[g]$.

This completes the proof of Theorem 4.

## B.3 Proof of Proposition 5

From the proof of Theorem 4 in Appendix B.2, we know that to obtain $\lambda_{\text{opt}} < 0$, it is sufficient to show

$$\mathbb{E}\frac{gh\zeta}{(1+\zeta)^3}\mathbb{E}\frac{\zeta}{(1+\zeta)^2} - \mathbb{E}\frac{\zeta^2}{(1+\zeta)^3}\mathbb{E}\frac{gh}{(1+\zeta)^2} - \frac{\tilde{\sigma}^2}{\gamma}\mathbb{E}\frac{\zeta^2}{(1+\zeta)^3} > 0.$$

With the distribution assumption on $(h, g)$, this is equivalent to the following

$$\gamma q(1-q)\frac{(h_1-1)(g_1-1)h_1 m^2}{(1+m)^3(1+h_1 m)^3} > \tilde{\sigma}^2\left((1-q)\frac{m^2}{(1+m)^3} + q\frac{h_1^2 m^2}{(1+h_1 m)^3}\right),$$

where $m = m(0)$ satisfies that

$$1 = \gamma\left((1-q)\frac{m}{1+m} + q\frac{h_1 m}{h_1 m + 1}\right).$$

(B.7)

This gives the following upper bound for $\tilde{\sigma}^2$:

$$\tilde{\sigma}^2 < \gamma q(1-q) \frac{(h_1-1)(g_1-1)h_1}{(1-q)(1+h_1 m)^3 + qh_1^2(1+m)^3}. \tag{B.8}$$

To provide a more intuitive result, we remove $m$ from (B.8). Note that from (B.7), we can derive the following straightforward upper bound for $m$:

$$m \leq \max\left(\frac{1}{h_1(\gamma q - 1)}, \frac{1}{\gamma - 1}\right),$$

which we plug in (B.8) and obtain

$$\tilde{\sigma}^2 < \gamma q(1-q) \max\left(\frac{(h_1-1)(g_1-1)h_1}{(1-q)\left(\frac{\gamma q}{\gamma q - 1}\right)^3 + qh_1^2\left(\frac{\gamma}{\gamma-1}\right)^3}, \frac{(h_1-1)(g_1-1)h_1(\gamma-1)^3}{(1-q)(h_1+\gamma-1)^3 + qh_1^2\gamma^3}\right).$$

### B.4 Proof of Proposition 6

Note that (3.1) holds for $\gamma < 1$ as well. It is clear that the bias term is non-negative and is strictly positive when $\lambda \neq 0$. Hence, we know the bias achieve its minimum only at $\lambda = 0$. We therefore only need to demonstrate that the variance term converges to a decreasing function of $\lambda$ for $\lambda > -c_0$.

Let $s(z)$ be the Stieltjes transform of the limiting distribution of the eigenvalues of $\mathbf{X}_{/w}^\top \mathbf{X}_{/w}$, then we have $s(-\lambda)$ satisfying

$$s(-\lambda) = \mathbb{E}\frac{1}{h(1 - \gamma + \gamma\lambda s(-\lambda)) + \lambda}. \tag{B.9}$$

In addition, from the Marchenko-Pastur law, the minimal eigenvalue of $\mathbf{X}_{/w}^\top \mathbf{X}_{/w}$ is bounded by $\inf_{v \in [c_l, c_u]} h \cdot (1 - \sqrt{\gamma})^2$. Hence, when $\gamma < 1$, for all $\lambda > -c_0$, we know that $s(-\lambda)$ is well defined and positive. Observe that for all $\lambda \neq 0$, $m(-\lambda)$ and $s(-\lambda)$ satisfies the following relation

$$m(-\lambda) = \frac{1-\gamma}{\lambda} + \gamma s(-\lambda). \tag{B.10}$$

Therefore from the proof of Theorem 1, we have the exact same expression of Part 1 for $\gamma < 1$ and $\lambda \neq 0$:

$$\text{Part 1} \xrightarrow{\text{p}} \tilde{\sigma}^2 \frac{m'(-\lambda)}{m^2(-\lambda)} \quad \forall \lambda > -c_0, \lambda \neq 0.$$

When $\lambda = 0$, we should replace $m(-\lambda)$ by $s(-\lambda)$ using (B.10). Since Part 1 is a continuous function of $\lambda$, we only need to focus on $\lambda \neq 0$ and show the following equation for all $\lambda > -c_0$ and $\lambda \neq 0$:

$$-\frac{2\gamma(m'(-\lambda))^2}{m^3(-\lambda)}\tilde{\sigma}^2 \frac{\mathbb{E}\frac{\zeta^2}{(1+\zeta)^3}}{1 - \gamma\mathbb{E}\frac{\zeta^2}{(1+\zeta)^2}} < 0. \tag{B.11}$$

Although we have proved (B.11) for the case $\gamma > 1$ in Appendix B.2, we used the fact that $m(-\lambda) > 0$ which is not guaranteed when $\gamma < 1$. In fact, only $s(z)$ and its any order derivatives are guaranteed to be positive on $z < c_0$, and $m(-\lambda)$ can be negative. Hence, we need to rederive (B.11) for $\gamma < 1$. From (B.5) and (B.4), what is left to be shown is that

$$\frac{m'(-\lambda)}{m(-\lambda)} \cdot \mathbb{E}\frac{\zeta^2}{(1+\zeta)^3} = m'(-\lambda) \cdot \mathbb{E}\frac{h^2 \cdot m(-\lambda)}{(1 + h \cdot m(-\lambda))^3} > 0, \quad \forall \lambda > -c_0, \lambda \neq 0. \tag{B.12}$$

By taking derivatives on both sides of (B.10) and from $s'(-\lambda) > 0$, we have

$$m'(-\lambda) - \frac{1-\gamma}{\lambda^2} = \gamma s'(-\lambda) > 0.$$

We therefore have $m'(-\lambda) > 0$ and (B.12) clearly holds when $m(-\lambda) > 0$. Since $\lambda > 0$ implies $m(-\lambda) > 0$ due to (B.10), we only need to show (B.12) when $\lambda < 0$ and $m(-\lambda) < 0$. We claim

that when $\lambda < 0$ and $m(-\lambda) < 0$, $1 + h \cdot m(-\lambda) < 0$ holds almost surely, and thus (B.12) is true due to

$$m(-\lambda) < 0 \quad \text{and} \quad \mathbb{E} \frac{h^2}{(1 + h \cdot m(-\lambda))^3} < 0.$$

We use contradiction to prove the claim. Suppose there exists $c_v > \inf h \triangleq c_h$ such that $1 + h \cdot m(-\lambda) > 0$ for all $h < c_v$ and the probability of $h < c_v$ is positive. Then let $c_m = -m(-\lambda)^{-1}$, we have $c_m > c_v > c_h > 0$. Furthermore, from (B.3) and definition of $c_0$, we have

$$-c_h(1 - \sqrt{\gamma})^2 = -c_0 < \lambda = -c_m - \gamma \mathbb{E} \frac{h}{h \cdot m(-\lambda) + 1} \leq -c_m + \gamma \mathbb{E} \frac{h \cdot c_m}{h - c_m} \mathbb{I}_{h > c_m}.$$

Therefore, we have

$$\mathbb{E} \frac{h}{h - c_m} \mathbb{I}_{h > c_m} > \frac{1}{\gamma} \left( 1 - \frac{c_h}{c_m} (1 - \sqrt{\gamma})^2 \right) > \frac{2 - \sqrt{\gamma}}{\sqrt{\gamma}} > 0. \tag{B.13}$$

On the other hand, since $m'(-\lambda) > 0$, from (B.4), we have

$$0 < \frac{1}{m^2(-\lambda)} - \gamma \mathbb{E} \frac{h^2}{(1 + h \cdot m(-\lambda))^2} \leq c_m^2 - \gamma \mathbb{E} \frac{h^2 c_m^2}{(h - c_m)^2} \mathbb{I}_{h > c_m},$$

which is equivalent to

$$\frac{1}{\gamma} > \mathbb{E} \left( \frac{h}{h - c_m} \mathbb{I}_{h > c_m} \right)^2.$$

However, from (B.13) and Jensen's inequality, we have

$$\mathbb{E} \left( \frac{h}{h - c_m} \mathbb{I}_{h > c_m} \right)^2 > \frac{(2 - \sqrt{\gamma})^2}{\gamma} > \frac{1}{\gamma}.$$

We have arrived at a contradiction and thus $1 + hm(-\lambda) < 0$ should hold almost surely when $\lambda < 0$ and $m(-\lambda) < 0$.

### B.5 Proof of Proposition 7

First note that in the setup of general data covariance and isotropic prior on $\boldsymbol{\beta}_*$, the prediction risk under optimal ridge regularization is given in [DW18, Theorem 2.1] as

$$R(\lambda_{\text{opt}}) = \frac{1}{\lambda_{\text{opt}} m(-\lambda_{\text{opt}})}, \tag{B.14}$$

where $\lambda_{\text{opt}} = \tilde{\sigma}^2 \gamma / c$. Note that (4.2) implies that $m(-\lambda_{\text{opt}})$ satisfies the following equation when $\gamma > 1$ or when $\gamma > 0$ and $\tilde{\sigma}^2 > 0$[9]:

$$\lambda_{\text{opt}} m(-\lambda_{\text{opt}}) = 1 - \gamma \mathbb{E} \frac{h \cdot m(-\lambda_{\text{opt}})}{1 + h \cdot m(-\lambda_{\text{opt}})}. \tag{B.15}$$

Therefore, taking the derivative of (B.14) with respect to $\gamma$ yields

$$\begin{aligned}
\frac{dR(\lambda_{\text{opt}})}{d\gamma} &\propto -\frac{1}{\gamma^2} \frac{1}{m(-\lambda_{\text{opt}})} + \frac{1}{\gamma} \frac{\partial m^{-1}(-\lambda_{\text{opt}})}{\partial \gamma} \\
&= -\frac{1}{\gamma^2} \frac{1}{m(-\lambda_{\text{opt}})} + \frac{1}{\gamma} \left( \frac{\tilde{\sigma}^2}{c} + \mathbb{E} \frac{h}{1 + h \cdot m(-\lambda_{\text{opt}})} + \gamma \frac{\partial}{\partial \gamma} \mathbb{E} \frac{h}{1 + h \cdot m(-\lambda_{\text{opt}})} \right) \\
&\propto \frac{\partial}{\partial \gamma} \mathbb{E} \frac{h}{1 + h \cdot m(-\lambda_{\text{opt}})} \\
&= -\left( \mathbb{E} \frac{h^2}{(1 + h \cdot m(-\lambda_{\text{opt}}))^2} \right) \frac{\partial m(-\lambda_{\text{opt}})}{\partial \gamma}. \tag{B.16}
\end{aligned}$$

[9]Note that when $\boldsymbol{\Sigma}_\beta = \boldsymbol{I}$ and $\tilde{\sigma}^2 > 0$ we have $\lambda_{\text{opt}} > 0$. Hence $m(-\lambda_{\text{opt}})$ exists for all $\gamma > 0$.

For notational convenience we define $\alpha = \tilde{\sigma}^2/c$, i.e. $\lambda_{\mathrm{opt}} = \alpha\gamma$. Note that for a fixed $\alpha$, $m(-\lambda_{\mathrm{opt}})$ is a function of $\gamma$. Thus we let $u(\gamma) \triangleq m(-\alpha\gamma)$. Then (B.16) implies that we only need to show $\frac{\mathrm{d}u(\gamma)}{\mathrm{d}\gamma} < 0$. Taking the derivative with respect to $\gamma$ on both sides of (B.15), we have

$$\alpha u(\gamma) + \alpha\gamma\frac{\mathrm{d}u(\gamma)}{\mathrm{d}\gamma} = -\mathbb{E}\frac{h \cdot u(\gamma)}{1 + h \cdot u(\gamma)} - \gamma\mathbb{E}\frac{h^2}{(1 + h \cdot u(\gamma))^2} \cdot \frac{\mathrm{d}u(\gamma)}{\mathrm{d}\gamma},$$

which is equivalent to

$$\begin{aligned}
\frac{\mathrm{d}u(\gamma)}{\mathrm{d}\gamma} &= \frac{-\alpha u(\gamma) - \mathbb{E}\frac{h \cdot u(\gamma)}{1 + h \cdot u(\gamma)}}{\alpha\gamma + \gamma\mathbb{E}\frac{h^2}{(1 + h \cdot u(\gamma))^2}} \\
&= \frac{-\alpha m(-\lambda_{\mathrm{opt}}) - \mathbb{E}\frac{h \cdot m(-\lambda_{\mathrm{opt}})}{1 + h \cdot m(-\lambda_{\mathrm{opt}})}}{\alpha\gamma + \gamma\mathbb{E}\frac{h^2}{(1 + h \cdot m(-\lambda_{\mathrm{opt}}))^2}} \\
&= -\frac{1}{\gamma\left(\alpha\gamma + \gamma\mathbb{E}\frac{h^2}{(1 + h \cdot m(-\lambda_{\mathrm{opt}}))^2}\right)},
\end{aligned}$$

where the last inequality holds due to (B.15). Finally, when $\tilde{\sigma}^2 = 0$ and $\gamma < 1$, we know that $\lambda_{\mathrm{opt}} = 0$ and $R(\lambda_{\mathrm{opt}}) = 0$. We thus know that the prediction risk $R$ is increasing as a function of $\gamma \in (0, \infty)$.

## C   Proofs omitted in Section 6

### C.1   Proof of Theorem 8

We first show that $\boldsymbol{\Sigma}_w = \boldsymbol{\Sigma}_x^{-1}$, i.e., $r$ being a point mass, is the optimal $\boldsymbol{\Sigma}_w$ for the variance term. From (B.3) and (B.4), we know the the variance function $\mathrm{R}_{\mathrm{v}}(r)$ can be written as

$$\mathrm{R}_{\mathrm{v}}(r) = \tilde{\sigma}^2\frac{1}{1 - \gamma\mathbb{E}\frac{\zeta_r^2}{(1+\zeta_r)^2}} = \tilde{\sigma}^2\frac{1}{\gamma\mathbb{E}\frac{\zeta_r}{(1+\zeta_r)^2}},$$

where we define $\zeta_r = r \cdot m(0)$ in this proof. Note that $\frac{\zeta_r}{1+\zeta_r}$ and $\frac{1}{1+\zeta_r}$ are both monotonic function of $\zeta_r$ with different monotonicity, we thus have

$$\mathbb{E}\frac{\zeta_r}{(1 + \zeta_r)^2} \leq \mathbb{E}\frac{\zeta_r}{1 + \zeta_r}\mathbb{E}\frac{1}{1 + \zeta_r} = \frac{1}{\gamma}\left(1 - \frac{1}{\gamma}\right),$$

where the last equality holds due to (B.3). The equality is achieved only when $r$ is a single point mass. Hence, we have

$$\mathrm{R}_{\mathrm{v}}(r) \geq \frac{\tilde{\sigma}^2}{(1 - \frac{1}{\gamma})} = \frac{\tilde{\sigma}^2\gamma}{\gamma - 1}.$$

The minimum variance is achieved when $r$ is a single point mass, i.e., $\boldsymbol{D}_{x/w} = \boldsymbol{I}$ and therefore, $\boldsymbol{\Sigma}_w = \boldsymbol{\Sigma}_x^{-1}$.

For the bias term, we first show that $r \stackrel{\mathrm{a.s.}}{=} sv$, i.e., $\boldsymbol{\Sigma}_w = \bar{\boldsymbol{\Sigma}}_\beta^{-1}$ is the optimal choice of $r$ for all non-negative random variable[10]. The result for $r \in \mathcal{S}_r$ immediately follows because as long as $r \in \mathcal{S}_r$, $\mathrm{R}_{\mathrm{b}}$ remains the same when we replace $v$ by $\mathbb{E}[v|s]$. Suppose $r \neq sv$ almost surely. Let us define $r_\alpha = \alpha \cdot sv + (1 - \alpha)r$ and consider the following bias function $\mathrm{R}_{\mathrm{b}}(\alpha)$:

$$\mathrm{R}_{\mathrm{b}}(\alpha) \triangleq \frac{m_\alpha'(0)}{m_\alpha^2(0)}\gamma\mathbb{E}\frac{sv}{(r_\alpha \cdot m_\alpha(0) + 1)^2},$$

where $m_\alpha(-\lambda), m_\alpha'(-\lambda) > 0$ satisfy that

$$\lambda = \frac{1}{m_\alpha(-\lambda)} - \gamma\mathbb{E}\frac{r_\alpha}{1 + r_\alpha \cdot m_\alpha(-\lambda)}$$

$$1 = \left(\frac{1}{m_\alpha^2(-\lambda)} - \gamma\mathbb{E}\frac{r_\alpha^2}{(r_\alpha \cdot m_\alpha(-\lambda)+1)^2}\right)m_\alpha'(-\lambda). \tag{C.1}$$

Note that $m_\alpha(z)$ is the Stieltjes transform of the limiting distribution of the eigenvalues of $\frac{1}{n}\boldsymbol{X}_\alpha\boldsymbol{X}_\alpha^\top$ where the covariance matrix of the rows of $\boldsymbol{X}_\alpha$ has its eigenvalues weakly converging to the random variable $r_\alpha$. Hence $m_\alpha(0) > 0$ and $m_\alpha'(0) > 0$ are well defined.

Our goal is to show that $1 \in \operatorname{argmin}_\alpha \mathsf{R}_\mathsf{b}(\alpha)$. We define $\zeta_\alpha = r_\alpha \cdot m_\alpha(0)$. Then from (4.2) and (C.1), we know that (B.3) and (B.4) hold with $\zeta$ replaced by $\zeta_\alpha$. Hence, we have

$$\mathsf{R}_\mathsf{b}(\alpha) = \frac{\gamma\mathbb{E}\frac{sv}{(\zeta_\alpha+1)^2}}{1 - \gamma\mathbb{E}\frac{\zeta_\alpha^2}{(\zeta_\alpha+1)^2}} = \frac{\mathbb{E}\frac{sv}{(\zeta_\alpha+1)^2}}{\mathbb{E}\frac{\zeta_\alpha}{(\zeta_\alpha+1)^2}},$$

where the last equality holds due to (B.3). By taking derivatives with respect to $\alpha$ in (B.3), we know that

$$\left.\frac{\partial m_\alpha(-\lambda)}{\partial\alpha}\right|_{\lambda=0} = \left.-\frac{\mathbb{E}\frac{(sv-r)m_\alpha^2(-\lambda)}{(1+r_\alpha\cdot m_\alpha(-\lambda))^2}}{\mathbb{E}\frac{r_\alpha\cdot m_\alpha(-\lambda)}{(1+r_\alpha\cdot m_\alpha(-\lambda))^2}}\right|_{\lambda=0} = -\frac{m_\alpha(0)\cdot\mathbb{E}\frac{\psi_\alpha-\zeta_\alpha}{(1+\zeta_\alpha)^2}}{(1-\alpha)\mathbb{E}\frac{\zeta_\alpha}{(1+\zeta_\alpha)^2}}, \tag{C.2}$$

where $\psi_\alpha = sv \cdot m_\alpha(0)$. With (C.2), we have

$$\frac{\mathrm{d}\,\mathsf{R}_\mathsf{b}(\alpha)}{\mathrm{d}\,\alpha} \overset{(i)}{\propto} -2\mathbb{E}\frac{\psi_\alpha(\psi_\alpha-\zeta_\alpha)}{(1+\zeta_\alpha)^3}\left(\mathbb{E}\frac{\zeta_\alpha}{(1+\zeta_\alpha)^2}\right)^2 + 2\mathbb{E}\frac{\psi_\alpha\zeta_\alpha}{(1+\zeta_\alpha)^3}\mathbb{E}\frac{(\psi_\alpha-\zeta_\alpha)}{(1+\zeta_\alpha)^2}\mathbb{E}\frac{\zeta_\alpha}{(1+\zeta_\alpha)^2}$$
$$-\mathbb{E}\frac{\zeta_\alpha}{(1+\zeta_\alpha)^2}\mathbb{E}\frac{(1-\zeta_\alpha)(\psi_\alpha-\zeta_\alpha)}{(1+\zeta_\alpha)^3}\mathbb{E}\frac{\psi_\alpha}{(1+\zeta_\alpha)^2} + \mathbb{E}\frac{(1-\zeta_\alpha)\zeta_\alpha}{(1+\zeta_\alpha)^3}\mathbb{E}\frac{\psi_\alpha-\zeta_\alpha}{(1+\zeta_\alpha)^2}\mathbb{E}\frac{\psi_\alpha}{(1+\zeta_\alpha)^2},$$
$$\tag{C.3}$$

where in equation (i) we omitted the following positive multiplicative scalar:

$$\left((1-\alpha)m_\alpha(0)\left(\mathbb{E}\frac{\zeta_\alpha}{(1+\zeta_\alpha)^2}\right)^3\right)^{-1}$$

We claim that the RHS of (C.3) is equivalent to the following

$$\underbrace{-2\,\mathbb{E}\frac{(\psi_\alpha-\zeta_\alpha)^2}{(1+\zeta_\alpha)^3}\left(\mathbb{E}\frac{\zeta_\alpha}{(1+\zeta_\alpha)^2}\right)^2}_{A} \underbrace{-2\left(\mathbb{E}\frac{\psi_\alpha-\zeta_\alpha}{(1+\zeta_\alpha)^2}\right)^2\mathbb{E}\frac{\zeta_\alpha^2}{(1+\zeta_\alpha)^3}}_{B} + \underbrace{4\mathbb{E}\frac{\zeta_\alpha(\psi_\alpha-\zeta_\alpha)}{(1+\zeta_\alpha)^3}\mathbb{E}\frac{\psi_\alpha-\zeta_\alpha}{(1+\zeta_\alpha)^2}\mathbb{E}\frac{\zeta_\alpha}{(1+\zeta_\alpha)^2}}_{C}.$$
$$\tag{C.4}$$

We apply the AM-GM inequality on the first two terms and obtain that $A + B \geq 2\sqrt{AB}$, and then apply Cauchy-Schwartz inequality on $\sqrt{AB}$ and obtain that $\sqrt{AB} \geq C$. Hence we know $-2(A + B - 2C) \leq 0$ and the equality is achieved only when $\psi_\alpha \overset{\text{a.s.}}{=} \zeta_\alpha$ which implies $\alpha = 1$ or both $\psi_\alpha$ and $\zeta_\alpha$ are single point mass. For the later case, we have $\frac{\mathrm{d}\,\mathsf{R}_\mathsf{b}(\alpha)}{\mathrm{d}\,\alpha} \equiv 0$ for all $\alpha$ and therefore $\alpha = 1$, i.e., $r \overset{\text{a.s.}}{=} sv$, is one of the minimum solutions. For the first case, we know $\mathsf{R}_\mathsf{b}(\alpha)$ is a strictly decreasing function of $\alpha$ and achieves its minimum at $\alpha = 1$ which is $r \overset{\text{a.s.}}{=} sv$. To show (C.4), we first simplify the first two terms in the RHS of (C.3).

$$-2\mathbb{E}\frac{\psi_\alpha(\psi_\alpha-\zeta_\alpha)}{(1+\zeta_\alpha)^3}\left(\mathbb{E}\frac{\zeta_\alpha}{(1+\zeta_\alpha)^2}\right)^2 + 2\mathbb{E}\frac{\psi_\alpha\zeta_\alpha}{(1+\zeta_\alpha)^3}\mathbb{E}\frac{(\psi_\alpha-\zeta_\alpha)}{(1+\zeta_\alpha)^2}\mathbb{E}\frac{\zeta_\alpha}{(1+\zeta_\alpha)^2}$$
$$= -2A - 2\mathbb{E}\frac{\zeta_\alpha(\psi_\alpha-\zeta_\alpha)}{(1+\zeta_\alpha)^3}\left(\mathbb{E}\frac{\zeta_\alpha}{(1+\zeta_\alpha)^2}\right)^2 + 4C + 2\mathbb{E}\frac{2\zeta_\alpha^2-\psi_\alpha\zeta_\alpha}{(1+\zeta_\alpha)^3}\mathbb{E}\frac{(\psi_\alpha-\zeta_\alpha)}{(1+\zeta_\alpha)^2}\mathbb{E}\frac{\zeta_\alpha}{(1+\zeta_\alpha)^2}$$
$$= -2A + 4C - 2\mathbb{E}\frac{\zeta_\alpha(\psi_\alpha-\zeta_\alpha)}{(1+\zeta_\alpha)^3}\mathbb{E}\frac{\zeta_\alpha}{(1+\zeta_\alpha)^2}\mathbb{E}\frac{\psi_\alpha}{(1+\zeta_\alpha)^2} + 2\mathbb{E}\frac{\zeta_\alpha^2}{(1+\zeta_\alpha)^3}\mathbb{E}\frac{(\psi_\alpha-\zeta_\alpha)}{(1+\zeta_\alpha)^2}\mathbb{E}\frac{\zeta_\alpha}{(1+\zeta_\alpha)^2}$$
$$= -2A - 2B + 4C$$
$$-2\mathbb{E}\frac{\zeta_\alpha(\psi_\alpha-\zeta_\alpha)}{(1+\zeta_\alpha)^3}\mathbb{E}\frac{\zeta_\alpha}{(1+\zeta_\alpha)^2}\mathbb{E}\frac{\psi_\alpha}{(1+\zeta_\alpha)^2} + 2\mathbb{E}\frac{\zeta_\alpha^2}{(1+\zeta_\alpha)^3}\mathbb{E}\frac{\psi_\alpha-\zeta_\alpha}{(1+\zeta_\alpha)^2}\mathbb{E}\frac{\psi_\alpha}{(1+\zeta_\alpha)^2}.$$

Similarly for the last two terms of (C.3),

$$\mathbb{E}\frac{(1-\zeta_\alpha)\zeta_\alpha}{(1+\zeta_\alpha)^3}\mathbb{E}\frac{\psi_\alpha-\zeta_\alpha}{(1+\zeta_\alpha)^2}\mathbb{E}\frac{\psi_\alpha}{(1+\zeta_\alpha)^2} - \mathbb{E}\frac{\zeta_\alpha}{(1+\zeta_\alpha)^2}\mathbb{E}\frac{(1-\zeta_\alpha)(\psi_\alpha-\zeta_\alpha)}{(1+\zeta_\alpha)^3}\mathbb{E}\frac{\psi_\alpha}{(1+\zeta_\alpha)^2}$$

$$= \mathbb{E}\frac{(1-\zeta_\alpha)\zeta_\alpha}{(1+\zeta_\alpha)^3}\mathbb{E}\frac{\psi_\alpha-\zeta_\alpha}{(1+\zeta_\alpha)^2}\mathbb{E}\frac{\psi_\alpha}{(1+\zeta_\alpha)^2}$$

$$-\mathbb{E}\frac{\zeta_\alpha}{(1+\zeta_\alpha)^2}\mathbb{E}\frac{(\psi_\alpha-\zeta_\alpha)}{(1+\zeta_\alpha)^2}\mathbb{E}\frac{\psi_\alpha}{(1+\zeta_\alpha)^2} + 2\mathbb{E}\frac{\zeta_\alpha}{(1+\zeta_\alpha)^2}\mathbb{E}\frac{\zeta_\alpha(\psi_\alpha-\zeta_\alpha)}{(1+\zeta_\alpha)^3}\mathbb{E}\frac{\psi_\alpha}{(1+\zeta_\alpha)^2}$$

$$= -2\mathbb{E}\frac{\zeta_\alpha^2}{(1+\zeta_\alpha)^3}\mathbb{E}\frac{\psi_\alpha-\zeta_\alpha}{(1+\zeta_\alpha)^2}\mathbb{E}\frac{\psi_\alpha}{(1+\zeta_\alpha)^2} + 2\mathbb{E}\frac{\zeta_\alpha}{(1+\zeta_\alpha)^2}\mathbb{E}\frac{\zeta_\alpha(\psi_\alpha-\zeta_\alpha)}{(1+\zeta_\alpha)^3}\mathbb{E}\frac{\psi_\alpha}{(1+\zeta_\alpha)^2}.$$

<div align="right">(C.6)</div>

Combine (C.5) and (C.6), we have (C.4) holds.

## C.2 Proof of Proposition 9

Since $\boldsymbol{\Sigma}_w = \bar{\boldsymbol{\Sigma}}_\beta^{-1}$ is the optimal choice for $\boldsymbol{\Sigma}_w \in \mathcal{H}_w$, we only need to prove this proposition in the case when $\boldsymbol{\Sigma}_w = (f_v(\boldsymbol{\Sigma}_x))^{-1}$.

Note that the proposition holds in the regime $\theta\gamma > 1$ due to the proof of Theorem 8 and Corollary 3. When $\theta\gamma < 1$, denote the quantile functions of $s$ and $\tilde{s} \triangleq \mathbb{E}[v|s] \cdot s$ as $Q_1$ and $Q_2$ respectively. We have

$$
\begin{aligned}
\mathbb{E}\big[sv \cdot \mathbb{I}_{s<Q_1(1-\theta)}\big] &= \mathbb{E}\big[\mathbb{E}[v|s] \cdot s\mathbb{I}_{s<Q_1(1-\theta)}\big]\\
&= \mathbb{E}\big[\tilde{s}\mathbb{I}_{s<Q_1(1-\theta),\tilde{s}<Q_2(1-\theta)}\big] + \mathbb{E}\big[\tilde{s}\mathbb{I}_{s<Q_1(1-\theta),\tilde{s}\geq Q_2(1-\theta)}\big]\\
&\geq \mathbb{E}\big[\tilde{s}\mathbb{I}_{s<Q_1(1-\theta),\tilde{s}<Q_2(1-\theta)}\big] + Q_2(1-\theta)\mathbb{P}(s<Q_1(1-\theta),\tilde{s}\geq Q_2(1-\theta))\\
&= \mathbb{E}\big[\tilde{s}\mathbb{I}_{s<Q_1(1-\theta),\tilde{s}<Q_2(1-\theta)}\big] + Q_2(1-\theta)\mathbb{P}(s\geq Q_1(1-\theta),\tilde{s}<Q_2(1-\theta))\\
&\geq \mathbb{E}\big[\tilde{s}\mathbb{I}_{s<Q_1(1-\theta),\tilde{s}<Q_2(1-\theta)}\big] + \mathbb{E}\big[\tilde{s}\mathbb{I}_{s\geq Q_1(1-\theta),\tilde{s}<Q_2(1-\theta)}\big]\\
&= \mathbb{E}\big[\tilde{s}\mathbb{I}_{\tilde{s}<Q_2(1-\theta)}\big].
\end{aligned}
$$

From Corollary 3, we know that the risk achieved by the PCR estimator is at least the same as that of a second PCR estimator where we replace $(\boldsymbol{\Sigma}_x, \boldsymbol{\Sigma}_\beta)$ by $(\boldsymbol{\Sigma}_x\boldsymbol{\Sigma}_w^{-1}, \boldsymbol{I})$. From [XH19], the optimal risk achieved by the PCR estimate for $\theta\gamma < 1$ in the second PCR problem is worse than the full model risk $R(\mathbb{E}[v|s] \cdot s, 0)$ which is the same risks achieved by the minimum $\|\hat{\boldsymbol{\beta}}\|_{\boldsymbol{\Sigma}_w}$ solution. Hence we know that the minimum $\|\hat{\boldsymbol{\beta}}\|_{\boldsymbol{\Sigma}_w}$ solution outperforms the PCR estimate for $\theta\gamma < 1$ as well.

## C.3 Proof of Theorem 10

From (B.3) and (B.4), we have the following equivalent formula for the risk function $R(r, \lambda)$:

$$R(r, \lambda) = \frac{\tilde{\sigma}^2 + \gamma\mathbb{E}\frac{sv}{(1+\zeta_r)^2}}{1 - \gamma\mathbb{E}\frac{\zeta_r^2}{(1+\zeta_r)^2}},$$

where we define $\zeta_r = r \cdot m_r(-\lambda)$ in this proof. We also know that $m_r(-\lambda)$ satisfies

$$1 = \lambda m_r(-\lambda) + \gamma\mathbb{E}\frac{\zeta_r}{1+\zeta_r}.$$

Let us first consider $r \in \mathcal{H}_r$. The result for $r \in \mathcal{S}_r$ immediately follows because as long as $r \in \mathcal{S}_r$, $R(r, \lambda)$ remains the same when we replace $v$ by $\mathbb{E}[v|s]$. We now apply similar proof strategy of Theorem 8 in Section C.1. Consider any $r \in \mathcal{H}_r$ with $r \neq sv$ almost surely. We define $r_\alpha = \alpha \cdot sv + (1-\alpha)r$ and consider the following risk function $R_\alpha(\lambda)$:

$$R_\alpha(\lambda) = \frac{\tilde{\sigma}^2 + \gamma\mathbb{E}\frac{sv}{(1+\zeta_\alpha(\lambda))^2}}{1 - \gamma\mathbb{E}\frac{\zeta_\alpha(\lambda)^2}{(1+\zeta_\alpha(\lambda))^2}},$$

where $\zeta_\alpha(\lambda) = r_\alpha \cdot m_\alpha(-\lambda)$, and $m_\alpha(-\lambda)$ satisfies that

$$\lambda = \frac{1}{m_\alpha(-\lambda)} - \gamma \mathbb{E} \frac{r_\alpha}{1 + r_\alpha \cdot m_\alpha(-\lambda)}$$

Note that $m_\alpha(z)$ is the Stieltjes transform of the limiting distribution of the eigenvalues of $\frac{1}{n} \boldsymbol{X}_\alpha \boldsymbol{X}_\alpha^\top$ where the covariance matrix of the rows of $\boldsymbol{X}_\alpha$ has its eigenvalues weakly converges to the random variable $r_\alpha$. We define $c_\alpha = -\inf_{x \in \mathcal{K}} x > 0$, in which

$$\mathcal{K} = \text{support of the limiting distribution of the eigenvalues of } \frac{1}{n} \boldsymbol{X}_\alpha \boldsymbol{X}_\alpha^\top.$$

We know that $m_\alpha(-\lambda) \geq 0$ and $m'_\alpha(-\lambda) > 0$ for all $\lambda > c_\alpha$ and from Section 4 of [SC95], we know that

$$\lim_{\lambda \to c_\alpha^+} \gamma \mathbb{E} \frac{\zeta_\alpha^2(\lambda)}{(1 + \zeta_\alpha(\lambda))^2} = 1.$$

Furthermore, $m_\alpha(-\lambda) \to 0$ as $\lambda \to \infty$. Hence we know that $\lambda_{\text{opt}}(\alpha) = \arg\min_\lambda R_\alpha(\lambda)$ exists[11], and by taking derivatives with respect to $\lambda$ for $R_\alpha(\lambda)$, it is clear that $\lambda_{\text{opt}}(\alpha)$ should satisfy that

$$\frac{\tilde{\sigma}^2 + \gamma \mathbb{E} \frac{sv}{(\zeta_\alpha + 1)^2}}{1 - \gamma \mathbb{E} \frac{\zeta_\alpha^2}{(1 + \zeta_\alpha)^2}} = \frac{\mathbb{E} \frac{sv \cdot \zeta_\alpha}{(1 + \zeta_\alpha)^3}}{\mathbb{E} \frac{\zeta_\alpha^2}{(1 + \zeta_\alpha)^3}}, \tag{C.7}$$

where we slightly abuse the notation and use $\zeta_\alpha$ as a shorthand for $\zeta_\alpha(\lambda_{\text{opt}}(\alpha))$. We now consider the following optimization problem:

$$\min_\alpha R_\alpha(\lambda_{\text{opt}}(\alpha)).$$

Our goal is to show that $1 \in \arg\min_\alpha R_\alpha(\lambda_{\text{opt}}(\alpha))$, from which we have

$$\min_\lambda R_{sv}(\lambda) = R_\alpha(\lambda_{\text{opt}}(\alpha))|_{\alpha=1} \leq R_\alpha(\lambda_{\text{opt}}(\alpha))|_{\alpha=0} = \min_\lambda R_r(\lambda).$$

Which informs us that the optimal $r$ for optimal weighted ridge regression is $r \overset{\text{a.s.}}{=} sv$.

Taking the derivatives of $R_\alpha(\lambda_{\text{opt}}(\alpha))$ with respect to $\alpha$ yields

$$\begin{aligned}
\frac{\mathrm{d}\, R_\alpha(\lambda_{\text{opt}}(\alpha))}{\mathrm{d}\, \alpha} &= \frac{\partial R_\alpha(\lambda)}{\partial m_\alpha(-\lambda)} \cdot \left( \frac{\partial m_\alpha(-\lambda)}{\partial \lambda} \cdot \frac{\mathrm{d}\, \lambda_{\text{opt}}(\alpha)}{\mathrm{d}\, \alpha} + \frac{\partial m_\alpha(-\lambda)}{\partial \alpha} \right)\Bigg|_{\lambda = \lambda_{\text{opt}}(\alpha)} + \frac{\partial R_\alpha(\lambda)}{\partial \alpha}\Bigg|_{\lambda = \lambda_{\text{opt}}(\alpha)} \\
&\overset{(i)}{=} \frac{\partial R_\alpha(\lambda)}{\partial \alpha}\Bigg|_{\lambda = \lambda_{\text{opt}}(\alpha)} \\
&\propto -\mathbb{E} \frac{sv(\psi_\alpha - \zeta_\alpha)}{(1 + \zeta_\alpha)^3} \left( 1 - \gamma \mathbb{E} \frac{\zeta_\alpha^2}{(1 + \zeta_\alpha)^2} \right) + \mathbb{E} \frac{\zeta_\alpha(\psi_\alpha - \zeta_\alpha)}{(1 + \zeta_\alpha)^3} \left( \tilde{\sigma}^2 + \gamma \mathbb{E} \frac{sv}{(\zeta_\alpha + 1)^2} \right),
\end{aligned}$$
(C.8)

where equality (i) holds due to $\frac{\partial R_\alpha(\lambda)}{\partial m_\alpha(-\lambda)}\Big|_{\lambda = \lambda_{\text{opt}}(\alpha)} = 0$, and we defined $\psi_\alpha = sv \cdot m_\alpha(-\lambda_{\text{opt}}(\alpha))$ in this proof. In addition, the multiplicative scalar omitted in the last equation is the following positive constant

$$\left( \frac{1 - \alpha}{2\gamma} \left( 1 - \gamma \mathbb{E} \frac{\zeta_\alpha^2}{(1 + \zeta_\alpha)^2} \right)^2 \right)^{-1}.$$

Combining (C.7) and (C.8) yields

$$\begin{aligned}
\frac{\mathrm{d}\, R_\alpha(\lambda_{\text{opt}}(\alpha))}{\mathrm{d}\, \alpha} \leq 0 \quad &\Leftrightarrow \quad \mathbb{E} \frac{\psi_\alpha \zeta_\alpha}{(1 + \zeta_\alpha)^3} \mathbb{E} \frac{\zeta_\alpha(\psi_\alpha - \zeta_\alpha)}{(1 + \zeta_\alpha)^3} \leq \mathbb{E} \frac{\psi_\alpha(\psi_\alpha - \zeta_\alpha)}{(1 + \zeta_\alpha)^3} \mathbb{E} \frac{\zeta_\alpha^2}{(1 + \zeta_\alpha)^3} \\
&\Leftrightarrow \quad \left( \mathbb{E} \frac{\zeta_\alpha(\psi_\alpha - \zeta_\alpha)}{(1 + \zeta_\alpha)^3} \right)^2 \leq \mathbb{E} \frac{(\psi_\alpha - \zeta_\alpha)^2}{(1 + \zeta_\alpha)^3} \mathbb{E} \frac{\zeta_\alpha^2}{(1 + \zeta_\alpha)^3}.
\end{aligned}$$

Where the last inequality holds for all $\alpha \leq 1$ by Cauchy-Schwartz. Therefore,

$$1 \in \arg\min_\alpha R_\alpha(\lambda_{\text{opt}}(\alpha)).$$

This completes the proof of the theorem.

# D    Auxiliaries

## D.1    Experiment Setup

We include the detailed constructions of $d_x$ and $d_\beta$ and figures mentioned in the main text. The values of $d_x$ and $d_\beta$ used in Figure 2 are constructed in the following way:

- **Discrete to discrete**: For $d_x$, we set each quarter of elements to be $1, 3, 5$ and $7$ respectively; For $d_\beta$, we set one forth elements to be $8$ and rest of the elements to be $1$.

- **Discrete to continuous**: For $d_x$, we set half of the elements to be $1$ and rest of the elements to be $8$; For $d_\beta$, we i.i.d. sample from $\mathrm{unif}([1, 8])$.

- **Continuous to continuous**: For $d_x$, we i.i.d. sample from $\mathrm{unif}([1, 5])$; For $d_\beta$, we i.i.d. sample from random variable $a = \min(u^2 + 1, 5)$ where $u \sim \mathcal{N}(0, 1)$.

- **Continuous to discrete**: For $d_x$, we i.i.d. sample from $\mathrm{unif}([1, 8])$; For $d_\beta$, we set half of the elements to be $1$ and rest of the elements to be $7$.

The values of $d_x$ and $d_\beta$ used in Figure 7 are constructed as:

- We construct $a$ and $b$ to be two independent Gaussian $\mathcal{N}(0, 1)$ random variables.

- **Left**: Let $(d_{x,i}, d_{\beta,i}) \overset{\text{i.i.d.}}{\sim} (s, v)$ where $s = |a| + 5$ and $v = (a + b/2)^2 + 1$. It is then straight-forward to show that $f_v(s) = \mathbb{E}[v|s] = (s - 5)^2 + 5/4$. Hence, the optimal $\boldsymbol{\Sigma}_w \in \mathcal{S}_w$ is $\boldsymbol{\Sigma}_w = \left((\boldsymbol{\Sigma}_x - 5\boldsymbol{I})^2 + 1.25\boldsymbol{I}\right)^{-1}$.

- **Right**: Let $(d_{x,i}, d_{\beta,i}) \overset{\text{i.i.d.}}{\sim} (s, v)$ where $s = |a|^{-1} + 2$ and $v = (a + b/2)^2 + 1$. It is then straightforward to show that $f_v(s) = \mathbb{E}[v|s] = \frac{1}{(s-2)^2} + 5/4$. Hence, the optimal $\boldsymbol{\Sigma}_w \in \mathcal{S}_w$ is $\boldsymbol{\Sigma}_w = \left((\boldsymbol{\Sigma}_x - 2\boldsymbol{I})^{-2} + 1.25\boldsymbol{I}\right)^{-1}$.

The covariances in Figure 9 are constructed as follow (we remark that the slightly different scaling is to ensure that the resulting risk for each choice is roughly of the same magnitude to be presented in one figure):

- **Aligned:** We construct $d_x$ to be three point masses $(a, b, c)$ with weights $(4/11, 4/11, 3/11)$, respectively. We choose $\kappa = 50$ and locate $a = 1$, $b = a/\kappa$ and $c = b/\kappa$. We set $\boldsymbol{\Sigma}_\beta = 18/5 \cdot \boldsymbol{\Sigma}_x$.

- **Misaligned:** We construct $d_x$ to be the same as the aligned case, and set $\boldsymbol{\Sigma}_\beta = 4/9 \cdot \boldsymbol{\Sigma}_x^{-1}$.

- **Other:** We construct $d_x$ and $d_\beta$ to be the sum of two vectors $d_1$ and $d_2$, both of which consists of two point masses. $d_1$ has its first $1/5$ entries to be $1$ and the rest $1/10$, and $d_2$ has its first $4/5$ entries to be $1$ and the rest $1/10$. We set $d_x = 2 \cdot d_1$ and $d_\beta = (d_1 + d_2)/3$,

## D.2 Additional Figures

(a) Aligned, SNR $\xi = 5$     (b) Misaligned, SNR $\xi = 5$     (c) Random, SNR $\xi = 5$

Figure 8: Finite sample prediction risk $\tilde{\mathbb{E}}(\tilde{y} - \tilde{\boldsymbol{x}}^\top \boldsymbol{\beta}_\star)^2$ (experiment) and the asymptotic risk $R(\lambda)$ (theory) against $\lambda$ for standard ridge regression ($\boldsymbol{\Sigma}_w = \boldsymbol{I}_d$) under label noise with SNR $\xi = 5$. We set $\gamma = 2$ and $(n,p) = (300, 600)$. 'dc' and 'ct' stand for for discrete and continuous distribution, respectively. We write 'aligned' if $\boldsymbol{d}_x$ and $\boldsymbol{d}_\beta$ have the same order, 'misaligned' for the reverse and 'random' for random order. Colors indicate different combinations of $\boldsymbol{d}_x$ and $\boldsymbol{d}_\beta$. Note that our derived risk $R(\lambda)$ matches the experimental values for all cases.

(a) ridgeless regression risk.          (b) PCR risk.

Figure 9: Comparison of the ridgeless regression estimator [HMRT19] and the PCR estimator. We set $n = 300$, $\gamma = 5$ and SNR=50. Observe that the ridgeless regression risk exhibits multiple peaks in the overparameterized regime due to the anisotropic covariances (especially when $\boldsymbol{d}_x$ and $\boldsymbol{d}_\beta$ are misaligned). In contrast, the PCR risk is largely decreasing with $\theta$, especially for the misaligned case, which agrees with Proposition 3. We remark that the PCR risk is not always monotone for $\theta\gamma > 1$, as illustrated by the blue curve.

(a) $\boldsymbol{d}_x \sim \mathrm{unif}([1,3])$.          (b) $\boldsymbol{d}_x \sim \frac{1}{2}\delta_1 + \frac{1}{2}\delta_3$.

Figure 10: We set $\boldsymbol{\Sigma}_w = \boldsymbol{I}$ and $\boldsymbol{\Sigma}_\beta = \boldsymbol{\Sigma}_x^\alpha$. As $\gamma$ increases from 1.1 to 4, we show the optimal value of $\lambda$ and the solid lines represents the noiseless case $\tilde{\sigma} = 0$ and the dashed lines represents the noisy case with a fixed SNR $\xi$. The solid green line shows the level of 0. We set the distribution of $\boldsymbol{d}_x$ to be (a): uniform on $[1,3]$; (b):two point masses on 1 and 3 with half and half probability.

## Footnotes

[7]We can exchange expectation and derivatives because $\left|\frac{\partial\frac{h}{1+hm}}{\partial m}\right| = \frac{h}{(1+hm)^2} < \sup h$ when $m(-\lambda) > 0$

[8]We take pseudo-inverse when $\lambda = 0$

[10]We do not require $r$ being bounded away from 0 and $\infty$ because we focus on the function $\mathrm{R}_{\mathrm{b}}$ directly.

[11] As $\lambda \to \infty$, the LHS of (C.7) remains finite and the RHS of (C.7) goes to infinity. On the other hand, as $\lambda \to c_\alpha^+$, the LHS of (C.7) goes to infinity and the RHS of (C.7) remains finite.