[Reviews · NeurIPS 2020]

Review 1

Summary and Contributions: This paper studies the linear regression within an over-parameterized regime. Both the covariates and the coefficients are assumed to have some nontrivial covariances. Overparameterisation has received much interest in recent years, since typical deep learning models are indeed over-parameterized. The main result of the work is a precise characterization of prediction risk E(y-x^T\hat \beta_\lambda)^2 in the proportional limit p/n\to \gamma\in (1,\infty), and analyzed the sign of the optimal parameter \lambda. In particular, optimal parameter can be negative. This differs drastically from the standard regression case.

Strengths: This work is theoretically sound, with some empirical evaluations. Over-parameterization is important for understanding many deep learning techniques. The model under study is still linear, so not directly applicable to deep learning, but it is more general than many existing studies, and hence it presents an interesting contribution to the community.

Weaknesses: The main limitation is that the study only presents the results in an asymptotic regime. It is not directly clear to the referee that how much insight can be shedded into practically important issues, either in a finite data / model, or beyond the linear model. Nonetheless, I am convince that the work is still important enough. A few minor comments that can be improved. 1) Figure 1 is actually not referenced in the main text. 2) page 2, it would be nice to give a reference to the statement on line 41. 3) I find Definition 2 hard to understand. Especially what is "the same order" ? Does it refer to entrywise of the same order or the vector norm of the same order. It would be better to have a more precise definition, preferrably with a forma. 4) line 241, change "conjuncture" to "conjecture". One last point is that some numerical results on more complicated models would greatly strengthen the study, e.g., on simple two-layer neural networks.

Correctness: the claims and method seem correct

Clarity: the paper is well written

Relation to Prior Work: The discussion with respect to prior work is sufficient.

Reproducibility: Yes

Additional Feedback: I thank the authors for the detailed response, which has addressed all my concerns satisfactorily.


Review 2

Summary and Contributions: The paper analyzes the generalization error in overparametrized linear regression with general quadratic norm regularization. The authors provide exact characterization of risk in the proportional asymptotic limit. They also discuss the optimal choice of regularization parameter and thoretically explain when $\lambda$ should be negative. Furthermore, they discuss the optimal choice of PD matrix that defines the regularization norm.

Strengths: Analyzing different problems in the proportional asymptotic limit has gained a lot of interest recently. Looking at different problems in this limit can theoretically explain interesting phenomena that we observe such as double descent or the need for negative regularization parameter. This paper extends the results of recent works to a more general setting with fewer restrictive assumptions, by allowing the data to have general covariance and the regularzation norm be defined with a general PD matrix. They build on the risk characterization to determine the optimal choice of $\lambda$ (or its sign) and optimal choice of matrix for the norm in regularization.

Weaknesses: The main issue I have with the paper is about the novelty of the results. The authors mention that previous work on linear regression is not as general as current work. In particular, they either only allow isotropic features or signal. This paper which is arXived about a month before the NeurIPS deadline seems to do both: [1] Emami, Melikasadat, et al. "Generalization error of generalized linear models in high dimensions." arXiv preprint arXiv:2005.00180 (2020). (I would refer to this paper as [1]). The results of this paper allow to characterize the exact generalization error in the same asymptotic limit for Guassian data with general covariance and any regularization, which includes the $\ell_2$ type regularzations considered here, as well as more general regularizations like general $\ell_p$ norms. Here are my understanding of the differences of the results of the two papers: - In [1] the authors allow for a Gaussian feature with any covariance matrix, whereas your paper allow non-Gaussina features so long as they have bounded 12th centered-moment. - In [1] the authors allow any regularization whereas your paper considers only $\ell_2$-type regularizations. - [1] the authors consider generalized linear models which have linear regression as a special case, whereas this work only considers linear regression. - Your paper provides a closed form formula for the generalization error, but in [1] the generalization error is given via a set of recursive equations which they call state evolution (SE). But it seems like getting exact formulas from SE is possible in certain simple cases like linear ridge regression as they do in the appendix. I guess they could do it for a general $\ell_2$ norm (but they have not). As such, I think the main novelty of this paper are the results about optimal choice of $\lambda$ and $\Sigma_w$. But I they might not be significant enough contribution for a NeurIPS paper. I believe at least a good comparison to the results of [1] is needed in this work.

Correctness: I did not check all the equations in the proof carefully, but at a glance, they seem to be correct. Furthermore, the theoretical results match the empirical results which corroborates correctness of the theory.

Clarity: The paper is very well written and clear. It is easy to follow

Relation to Prior Work: Yes. The author makes it very clear what the main contributions of the paper are, and in what sense the results are more general compared to previous results.

Reproducibility: Yes

Additional Feedback: Minor issue: - Reference [EM75] is missing the authors. Post author feedback: After reading other reviews and the author feedback I still believe that my initial rating was correct.


Review 3

Summary and Contributions: This paper studies a weighted version of ridge regularization in overparameterized linear regression. Instead of penalizing the empirical risk by the usual $\ell_2$ norm, authors consider a general setup where the penalization belongs to a class of norms where it turns out that in several setups the euclidian norm is not optimal. Among other findings, this setup allows to elucidate the mystery behind negative optimal regularization that was observed lately in Neural Networks. This can be seen as one of the highlights of this paper.

Strengths: This paper considers a general setup of regularized linear regression where, unlike prior work, the design, signal and weighting matrix may be anisotropic. Using the notion of misalignment between signal and design, authors were moreover able to give theoretical justification to recent empirical phenomena in Neural Networks. For all these reasons I think that the present submission is relevant to the NeurIPS community.

Weaknesses: * Assumption 2 of codiagonalizability is pretty restrictive in terms of the class of allowed weighting matrices. * Optimal choices of the weighting matrix $\Sigma_w$ depend either on the unknown signal or the population covariance. While authors try to address this point in section 6 claiming that $\Sigma_x$ can be estimated from data, I think this does not entirely solve the problem. First what they propose requires a prior knowledge of the limiting spectral distributions. Second the limiting distribution of the empirical covariance of the design is singular and hence does not satisfy Assumption 1 in the overparameterized regime. It would be interesting to understand to which extent this theory is valid when using covariance estimators as weighting matrices.

Correctness: The claims and their proofs seem correct to me.

Clarity: Yes.

Relation to Prior Work: Yes.

Reproducibility: Yes

Additional Feedback: Comment: * In my opinion the main concern of the present paper is the adaptive choice of the optimal weighting matrix. I understand that it is not necessarily easy to achieve this task. I think a first step could be just to come up with a decision rule (independent from the signal) choosing the optimal weighting matrix between identity matrix and the covariance $\Sigma_x$. Another option is proving that interpolating between those two matrices may outperform both of them. Typos: * line -1 in the caption of Figure 1: "compared" ----------------------------------------------------------------------- Given the page restriction, I think the reply to my concerns was OK. I decided to keep my initial score which I believe is good for this submission.

[Author Response · NeurIPS 2020]

We thank all reviewers for their helpful feedback. Below we address the questions and comments individually.

**R1,R2,R3 - Writing.** We will correct typos in the main text and bibliography, and refer to Figure 1 in the introduction.

**R1 - Finite Data Analysis.** We remark that: (i) while our results are asymptotic in nature, our experiments showed
that they give accurate description of the prediction risk for dataset with moderate size, as demonstrated in Figure 2
($n = 300$); (ii) we believe that similar to previous works on double descent, our asymptotic characterization can be
translated to non-asymptotic guarantees using standard concentration tools.

**R1 - Definition of "Same Order".** We apologize for the confusion. As we specify a joint relation between covariances,
we use the term "order" to describe the increasing or decreasing trend of a vector, i.e. "same order" of $a$ and $b$ implies
$a_i \geq a_j$ iff $b_i \geq b_j$ for all i,j. When $\Sigma_x$, $\Sigma_\beta$ are codiagonalizable, "same order" of eigenvalues suggests that the features
are informative (learning is "easy"); this is analogous to a fast eigenvalue decay of the RKHS "source condition".

**R1 - Extension to Neural Network.** We believe that our analysis can be extended to neural nets that are well-described
by a kernel or random features model (which is linear regression under different features). This includes a two-layer
network with fixed 1st layer (RF model), or with trained 1st layer under overparameterization (the NTK model).

**R2 - Novelty & Comparison with [1].** We thank the reviewer for mentioning this reference – we were not aware of
this work (which is arxived in May) at the time of submission. However, we believe that the reviewer has misjudged the
relation and overlap between [1] and our work. We first note that computing the prediction risk is only a small part of
our contribution; we also provided precise characterization of the optimal ridge parameter and weighting matrix. More
importantly, the setup and theory in [1] are quite different than and *do not* produce our results. In particular:

• **[1] may not cover the case when $\lambda \leq 0$.** When the regularization strength $\lambda$ is negative, it is not guaranteed that
limit of VAMP exists due to non-convexity of the objective in the overparameterized regime. Even if we assume
VAMP converges for all $\lambda$ and estimation of generalization error is accurate, it is still not clear that VAMP converges
to the specific ridge solution studied in our paper when $\lambda$ is negative or 0 (which corresponds to minimum $\ell_2$
norm solution), since it is possible that the SE has multiple fixed points when $\lambda \leq 0$. We remark that establishing
convergence and uniqueness can also be challenging for analysis based on the Convex Gordon Min-max Theorem.

• **The VAMP framework does not capture our "aligned" or "misaligned" cases.** The key assumption in the VAMP
analysis in [1] is that the limiting distribution of $\beta_i^\star$, the components of the true signal, is independent to that of the
features. As a result, random permutation of $\beta_i^\star$ does not effect the generalization error of the VAMP estimate. This
corresponds to the "random order" case in our setup, for which we showed that the corresponding optimal ridge
parameter is always non-negative[1]. In contrast, we specified a general joint distribution between the eigenvalues of
$\Sigma_x$ and the components of $d_\beta$ (See exact definition in paper). As demonstrated in Figure 2, different joint distribution
leads to different generalization error even when the limiting spectral distributions of $\Sigma_x$ and $\Sigma_\beta$ are the same. It is
precisely this extension that enables us to explain the "negative ridge" phenomenon.

• **The analysis in [1] alone cannot characterize the optimal ridge parameter and weighting matrix.** As mentioned
above, the optimal ridge parameter is always non-negative under the assumption in VAMP (it is also not true that SE
can always be simplified to exact expressions.). Furthermore, we decided the optimal weighting matrix for optimal $\lambda$
or the bias and variance term separately, and also covered the case when it only depends on feature covariance.

**R3 - Optimal Choice of Weighting Matrix.** We provide the following clarification and comments on the optimal $\Sigma_w$.

• The data covariance $\Sigma_x$ considered in Section 6 is the population covariance matrix, *not* the empirical covariance
(which is degenerate when $\gamma > 1$). While it can be difficult to determine $\Sigma_x$ from labeled data alone, the quantity
can be estimated from additional unlabeled data (i.e., a semi-supervised setting[2]). We agree with the reviewer that
designing a weighting matrix solely based on the empirical covariance is an interesting problem.

• We agree that interpolating between weighting matrices is an intuitive strategy: Figure 4 and 5 consider various
powers of data covariance ($\Sigma_x^\alpha$ for different $\alpha$), which can be seen as a geometric interpolation between $I_p$ and $\Sigma_x$.

• As mentioned in the paragraph starting from line 251, given the data covariance, a reasonable decision rule is to
construct the weighting matrix based on certain polynomial transformations to $\Sigma_x$, the parameter of which can be
tuned via cross validation. Figure 5 and 8 show that such approach does indeed outperform standard ridge regression.

**R3 - On Assumption 2.** We agree with the reviewer that assumption 2 does not cover all possible weighting matrices.
We remark that this assumption is *not* needed in our risk calculation, but only in the characterization of optimal
weighting matrix (for convenient formulas). In addition, it is worth noting that similar codiagonalizability assumptions
is not uncommon in the theoretical study of regression models. For instance, the standard source condition in RKHS
regression is analogous to codiagonalizable $\Sigma_x$ and $\Sigma_\beta$ with certain eigenvalue decay in our setup.

## Footnotes

[1]This partially justifies the restriction to non-negative $\lambda$ in the VAMP framework.

[2]For example see: Tony Cai, T., and Zijian Guo. "Semisupervised inference for explained variance in high dimensional linear regression and its applications." Journal of the Royal Statistical Society: Series B.


[Meta-Review · NeurIPS 2020]

The paper received three positive reviews. Most of the minor concerns raised in the initial reviews have been addressed in the rebuttal. The area chair agrees with the reviewers' assessment and follows their recommendation.